

# River profile response to normal fault growth and linkage: An example from the Hellenic forearc of south-central Crete, Greece

Sean F. Gallen[1], Karl W. Wegmann[2]

[1]Geological Institute, Swiss Federal Institute of Technology (ETH), Zürich, Switzerland, 8092
[2] Department of Marine, Earth and Atmospheric Sciences, North Carolina State University, Raleigh, NC, USA, 27695

*Correspondence to*: Sean F. Gallen (sean.gallen@erdw.ethz.ch)

**Abstract.** Topography is a reflection of the tectonic and geodynamic processes that act to uplift the Earth's surface and the erosional processes that work to return it to base level. Numerous studies have shown that topography is a sensitive recorder or tectonic signals. A quasi-physical understanding of the relationship between river incision and rock uplift has made the

analysis of fluvial topography a popular technique for deciphering relative, and some argue absolute, histories of rock uplift. Here we present results from a study of the fluvial topography from south-central Crete demonstrating that river longitudinal profiles indeed record the relative history of uplift, but several other processes make it difficult to recover quantitative uplift histories. Prior research demonstrates that the south-central coastline of Crete is bound by a large (~100 km long) E-W striking composite normal fault system. Marine terraces reveal that it is uplifting between $0.1 – 1.0$ mm yr$^{-1}$. These studies

suggest that two normal fault systems, the offshore Ptolemy and onshore South-Central Crete faults linked together in the recent geologic past (Ca. $0.4 – 1$ Myrs bp). Fault mechanics predicts that when adjacent faults link into a single fault the uplift rate in the linkage zone will increase rapidly. Using river profile analysis we show that rivers in south-central Crete record the relative uplift history of fault growth and linkage, as theory predicts that they should. Calibration of the commonly used stream power incision model shows that the slope exponent, n, is ~ 0.5, contrary to most studies that find n ≥ 1.

Analysis of fluvial knickpoints shows that migration distances are not proportional to upstream contributing drainage area, as predicted by the stream power incision model. Maps of the transformed stream distance variable, χ, indicate that drainage basin instability, drainage divide migration and river capture events complicate river profile analysis in south-central Crete. Waterfalls are observed in southern Crete and appear to operate under less efficient and different incision mechanics than assumed by the stream power incision model. Drainage area exchange and waterfall formation are argued to obscure

linkages between empirically derived metrics and quasi-physical descriptions of river incision, making is difficult to quantitatively interpret rock uplift histories from river profiles in this setting. Karst hydrology, break down of assumed drainage area-discharge scaling and chemical weathering might also contribute to the failure of the stream power incision model to adequately predict the behavior of the fluvial system in south-central Crete.




# 1 Introduction

It has long been argued that landscapes are archives of spatial and temporal trends of tectonic activity (Davis, 1899; Penck and Penck, 1924). Theoretical advances and empirical datasets have furthered our understanding of how rivers react to changing boundary conditions (Howard et al., 1994; Whipple and Tucker, 1999, Tucker and Whipple, 2002). Through these studies it has been shown that landscapes adjust to tectonic, climate or topological (e.g. stream capture) perturbations over $10^4 - 10^6$ yr time scales (Snyder et al., 2000; Crosby and Whipple, 2006; Berlin and Anderson, 2007; Hilley and Arrowsmith, 2008; Val et al., 2014; Willett et al., 2014). Bedrock river systems have been shown to be particularly sensitive indicators of changing boundary conditions, define the relief structure in mountains (Whipple et al., 1999), and transmit signals of base level fall upstream throughout the landscape (Merrits and Vincient, 1989; Pazzaglia et al., 1998; Whipple and Tucker, 1999; Snyder et al., 2000). Transient landscapes, those landscapes in the process of adjusting to newly imposed boundary conditions, provide the opportunity to interrogate topography for information regarding the timing and magnitude of landscape adjustment to external forcing. As such, studies of transient landscapes have received considerable attention over the past 15 years and serve as a cornerstone of modern tectonic geomorphology studies (Kirby and Whipple, 2001, 2012; Wobus et al., 2006; Pritchard et al., 2009; Perron and Royden, 2013; Royden and Perron, 2013; Gallen et al., 2013; Goren et al., 2014). Most studies that seek to extract tectonic and climate signals from fluvial landscapes rely on river profile analysis; the interpretation of the geometry of river longitudinal profiles, mostly in the context of the detachment-limited river incision model (Howard et al., 1994; Whipple and Tucker, 1999). Beyond the river basin(s) being studied, understanding the response of erosional landscapes to tectonic, climate or topological perturbations is essential for interpreting stratigraphic records in sedimentary basins (e.g. Armitage et al., 2011), understanding the timescales of orogenic buildup and decay (Whipple, 2009), and is a compliment to deciphering the timing of molecular genetic variations in aquatic populations (e.g., Montgomery, 2000; Craw et al., 2008; Mackey et al., 2011).

The proposed role of bedrock rivers as a governor on landscape evolution in erosional settings has led to a concerted effort by the geomorphic community to identify, document and quantify controls on long-term bedrock river incision (Howard and Kerby, 1983; Howard et al., 1994; Whipple and Tucker, 1999; Sklar and Dietrich, 2004; Whipple, 2004; Whipple et al., 2013; Lague, 2014). Our best understanding of controls on long-term bedrock river incision has mostly been gleaned from studies of landscapes thought to be at or near steady-state conditions, where rates of local rock uplift and erosion are balanced at geologically meaningful time scales ($> 10^4$ yrs) (Ouimet et al., 2008; DiBiase et al., 2010). These studies illustrate a systematic and monotonic increase in erosion rate with metrics based on stream power or shear-stress models or river incision, but it is important to recognize that many different river incision models (detachment-limited, transport-limited, or sediment-flux dependent) make similar predictions of river morphology and erosion rate at steady-state (Tucker and Whipple, 2002; Tucker, 2004; Gasparini et al., 2006). However, studies of river incision in transient landscapes show a much more complicated story where rivers exhibit changes in channel geometry, sediment flux and grain size, and the amount of bed cover and roughness that singularly or collectively may affect local rates of incision (Sklar and Dietrich,



2004; Finnegan et al., 2005, 2007; Amos and Burbank, 2007; Whittaker et al., 2007; Cook et al., 2013; Attal et al., 2015; Shobe et al., 2016). These responses to transient river incision will affect the pace and style of landscape adjustment to a base level fall event, as predicted by different classes of incision models (Tucker and Whipple, 2002; Tucker, 2004; Gasparini et al., 2006).

Despite the ability of existing river incision models to capture some of the range of the observed behavior of transient river systems, they often fail to describe some important and commonly observed aspects of such systems. In order to make the physical world more tractable, most incision models assume that (1) sediment flux scales with drainage area (Sklar and Dietrich, 2004; Lamb et al., 2007), that (2) the incision process in oversteepened reaches (e.g. waterfalls) is the same as in lower gradient reaches, and (3) that contributing drainage area remains fixed in space and time. Recent studies by

Attal et al. (2015) and Shobe et al. (2016) explore the role that spatially variable sediment flux and caliber has on incision processes in transient landscapes. These studies highlight observations of spatially heterogeneous changes in the flux and size of sediment delivered to the channel that might be missed by simple expressions of sediment flux that are only scaled to local channel slope and drainage area, a proxy for discharge. For example, Shobe et al. (2016) suggested that enhanced delivery of locally-derived hillslope boulders to a channel can impact bed cover and shear stresses such that river incision

can be slowed in steeper reaches, counter to the assumptions of a straightforward stream power approach to determining incision rates. Brocard et al. (2016) argue that in regions with highly weathered uplands, where only fine grained sediment is produced during surface erosion, knickpoint migration efficiency may be reduced, resulting in downstream reaches over-steepened due to a deficiency of bedload tools.

      When river reaches become over-steepened and waterfalls form, the relative importance of incision mechanisms

assumed by a stream-power based incision model, such as abrasion and plucking (e.g. Whipple et al., 2000), give way to processes such as plunge pool drilling, slope undercutting and rock toppling (Lamb, et al., 2007; Lamb and Dietrich, 2009; Haviv et al., 2010). It has been argued that these over-steepened sections of rivers may act to enhance or inhibit river incision, as well as to obscure tectonic and climatic information that may be encoded in river systems (Haviv et al., 2006; Lamb et al., 2007). For example, DiBiase et al. (2015) showed that the development of waterfalls associated with transient

slope-break knickpoints (Kirby and Whipple, 2012) related to increases in uplift rate observed along Big Tujunga creek in the San Gabriel Mountains of California generally acted to slow knickpoint retreat rates at higher elevations in the catchment relative to predictions from detachment-limited stream power models, while in contrast some low elevation waterfalls appear to travel faster than theoretical expectations.

      River networks are not static topographic features. Instead, they reshape themselves based on changing boundary

conditions until gravitationally-driven equilibrium is achieved between conjoined catchments (Bishop et al., 1995; Prince et al., 2010; 2011; Willett et al., 2014). This spatial rearrangement is driven by slow progressive divide migration and by discrete river capture events (Prince et al., 2010; 2011; Val et al., 2014; Willett et al., 2014). Most analyses of river networks however operate under the implicit assumption that drainage area is fixed in space and time. In regions with variable climate



and changing uplift rates it is reasonable to assume that drainage network patterns will adjust to attain a new equilibrium (Willett et al., 2014). Drainage area exchange between adjacent catchments during adjustment to new boundary conditions will affect the basin response time and fluvial incision rates. This concept is best understood in the context of river incision models that assume incision rate is proportional to drainage area, in which many of the parameters used to model incision are

assumed to scale in some way with drainage area (e.g. discharge, sediment flux, river width, local channel slope). All else being equal, a loss of drainage area will increase response times and decrease river incision rates; a gain in drainage area will reduce response times and accelerate river incision rates. These changes in drainage area complicate interpretations of changes in tectonic and climate signals preserved in river networks because river profile analysis operates under the implicit assumption that the drainage network has remained static over time.

In this study we seek to determine factors that control the evolution of mountain streams along a rapidly uplifting coastline from the island of Crete, Greece, and to assess the nature in which these streams record tectonic signals and how well they conform to predictions of stream-power incision models (c.f. Tomkins et al., 2003). The study site consists of 21 catchments that drain the footwalls of active extensional faults that form the range front of the Asterousia and Dikti mountains (Fig. 1). Previous studies of Pleistocene marine terraces document mid-to-late Quaternary uplift rates along the

coastline in the Asterousia Mountains, demonstrating that the normal faults that bound these two ranges linked prior to the mid Pleistocene (Gallen et al., 2014). Topographic evidence of a long-term slip-deficit in the linkage zone between the two fault systems suggests that linkage occurred in the recent geologic past (Gallen et al., 2014). We exploit fault mechanics theory that predicts a rapid increase in the rate of uplift in the vicinity of a fault linkage to establish a relative uplift history for the study area; first an early phase of uplift consistent with growth of two isolated fault systems followed by a later phase

related to the linkage and development of a new composite fault system (Gallen et al., 2014). We then use this natural experiment to assess the fluvial response to this step-change in uplift rate across the fault linkage zone. We preform analytical and statistical analyses of river profile segments and knickpoints to assess the ability of the detachment-limited river incision model to predict observations. We discuss the potential influence of lithology, waterfalls and drainage area exchange on the evolution of these river systems with broader implications for extracting tectonic and climate signals from

fluvial topography.

## 2 Background

### 2.2 Geologic and Tectonic setting of Crete

The island of Crete occupies a rapidly extending and uplifting forearc high above the Hellenic Subduction zone (Angelier et al., 1982; Gallen et al., 2014). The Hellenic subduction zone currently accommodates ~ 35 mm yr$^{-1}$ of motion

between the down-going African and overriding Eurasian plates (McClusky et al., 2000; Reilinger et al., 2006). A well-defined Benioff seismic zone illuminates the Hellenic subduction interface as it dips northward at 10° to 15°, reaching depths





of 35 to 45 km beneath Crete (Papazachos et al., 1996, 2000; Knapmeyer, 1999). The crust underlying Crete is composed of compressional nappes that were emplaced between about 35 to 15 Ma (Jolivet and Brun, 2010 and references therein). Some of the nappe units were subducted and underplated between 24 to 19 Ma (Theye and Seidel, 1993) and rapidly exhumed by ~ 18 to 16 Ma (Thomson et al., 1998; Brix et al., 2002; Rahl et al., 2005; Marsellos et al., 2010). Southward extension of the

Aegean domain initiated at approximately 23 Ma, and since the Miocene horizontal brittle extension has dominated deformation in the upper crust (Angelier et al., 1982; Bohnhoff et al., 2001; Fassoulas 2001; van Hinsbergen and Schmid, 2012). Brittle extension continues today in the vicinity of Crete as noted by numerous active roughly N-S and E-W striking faults (Fassoulas et al., 1994; Bohnhoff et al., 2005; Caputo et al., 2006; Mouslopoulou et al., 2011, 2014; Gallen et al., 2014). On Crete, extensional faults cut the structural nappe units and opened basins that filled with Neogene marine

sediments that are now exposed 100's of meters above sea-level, indicating long-lived uplift of the island (Meulenkamp et al., 1994; van Hinsbergen and Meulenkamp, 2006; Zachariasse et al., 2008). Pleistocene and Holocene paleo-shoreline markers found 10's to 100's of meters above modern sea level along the coastlines of Crete document continued uplift of the island throughout the Quaternary (Flemming, 1978; Angelier, 1979; Pirazzoli et al., 1982; Meulenkamp et al., 1994; Kelletat, 1996; Wegmann, 2008; Strasser et al., 2011; Strobl et al., 2014; Gallen et al., 2014; Tiberti et al., 2015;

Mouslopoulou et al., 2015).

## 2.2 Tectonic geomorphology of south-central Crete

The Asterousia and Dikti mountains that currently define the south-central coastline of Crete are horst structures embedded in the Hellenic forearc high (Fig 1). The horsts are primarily composed of Mesozoic carbonates and interbedded mud and sandstones with lesser amounts of ophiolite and Neogene turbidites (Fig. 1). Gallen et al. (2014) mapped

Pleistocene marine terraces and used optically stimulated luminesce (OSL) dating of deposits and correlations to the mid-to-late Quaternary sea level curve to derive rock uplift histories across an ~ 100 km stretch of the Cretan coastline (Fig. 1). The authors showed that the marine terraces are deformed and cut by two major extensional faults that extend onshore near the towns of Lentas and Tsoutsouros (Fig. 1). The eastern portion of the Asterousia Mountains is uplifting at eastward-increasing rates between 0.4-0.8 m kyr$^{-1}$ during the Pleistocene as part of the footwall of an offshore normal fault, known as

the Ptolemy fault (Fig. 1). At the town of Tsoutsouros, the marine terraces are truncated by a major normal fault that extends on shore, known as the South-Central Crete fault (SCCF), which represents the onshore continuation of the Ptolemy fault (Fig. 1). East of Tsoutsouros the SCCF forms the onshore segmented range front of the Dikti Mountains. Pleistocene marine terraces offset by the SCCF at Tsoutsouros provide long-term average slip rates of ~ 0.35 m kyr$^{-1}$, while the terraces cut into the hanging wall of the SCCF record uplift rates of 0.1 – 0.3 m kyr$^{-1}$ (Gallen et al., 2014). The observation that terraces are

uplifted in the hanging wall of the SCCF is of regional importance and signifies that geodynamic processes responsible for regional uplift of the island of Crete are outpacing upper crustal thinning accommodated by motion on active extensional faults. Gallen et al. (2014) further interpret the Ptolemy and SCCF faults as previously independent growing fault systems that linked in the geologically recent past (< 1 Ma) and that the Tsoutsouros area represents the linkage zone between the





two fault systems. This interpretation is based on the observations that the long-term displacement history inferred from swath topographic profiles that are parallel to the footwall mountain ranges are consistent with displacement on two isolated fault systems (Fig. 1a, c), while marine terraces that are inferred to have formed at ~ ≤ 400 kyrs are now offset by a single linked fault (Fig. 1d).

## 2.3 Climatic and Geomorphic setting

The island of Crete is in the Mediterranean climate zone, with cool moist winters and warm dry summers. Mean annual precipitation is ~ 650 mm yr$^{-1}$, locally increasing to > 1000 mm in the mountain ranges due to orographic lifting (Rackham and Moody, 1996). Average temperatures are 27° C in summer and 12° C in winter, and winds prevail from the north to northwest year-round, but are strongest during the summer months. Climate in the Eastern Mediterranean is generally drier during glacial or stadial intervals relative to the present-day, whereas interstadials are thought to be analogous to today's climate (e.g., Tzedakis, 2009). Sediment production is inferred to be higher relative to the modern during much of the Late Quaternary, especially during climatic transitions that force variations in hillslope vegetation communities, as evidenced by available geochronology for large alluvial fans along the south coast (e.g., Pope et al., 2008, 2016; Wegmann, 2008; Gallen et al., 2014).

Most of the rivers draining the Asterousia and Dikti Mountains are ephemeral. The geomorphic system in South-Central Crete is presently supply-limited; most channel beds have a thin alluvial cover with local patches of exposed bedrock and a general absence of floodplains and Holocene terraces, except along the lower coastal zones where larger rivers discharge into the sea. Where rivers cut across the Neogene sedimentary and Mesozoic sedimentary and ophiolite units the dominant mode of river incision occurs by physical processes such as abrasion and plucking of the underlying bedrock (e.g., Whipple et al., 2000, 2013; Whipple, 2004). Valleys cut into these units are generally wide and V-shaped, hillslopes are at or near threshold gradients, and hillslope sediment transport is primarily by shallow landsliding. In contrast, hillslopes underlain by Mesozoic carbonates exhibit shallow landsliding of colluvial material detached from the local bedrock by dissolution and/or physical weathering. Rock falls are common where locally-steepened carbonate hillslope and river canyon reaches exist. Chemical dissolution appears to be an important mass-transport process in the carbonate units, evidenced by the development of narrow slot canyons and karst drainage observed in many places throughout Crete.

## 2.4 Relationships between fault growth, linkage and footwall uplift patterns

Fault system properties including length, width, and displacement often follow quasi-universal scaling relationships (Walsh and Watterson, 1988; Cowie and Scholz, 1992b; Dawers et al., 1993; Nicol et al., 1996; Schlische et al., 1996; Gupta and Scholz, 2000). For example, globally compiled datasets of fault length (L) and maximum fault displacement ($D_{max}$) exhibit a linear relationship between these two parameters where $D_{max}$ is consistently ~ 3% of fault length (Walsh and Watterson, 1988; Cowie and Scholz, 1992a,b; Dawers et al., 1993; Schlische et al., 1996; Manighetti et al., 2001). These



findings have led to models whereby single isolated faults grow through repeated earthquakes in a self-similar fashion; propagating laterally at their tips while accumulating displacement maxima near their centers (Drawers et al., 1993). Displacement profiles often reveal the history of fault growth and linkage for active fault zones (Fig. 2; Anders and Schlische, 1994; Dawers and Anders, 1995; Gupta and Scholz, 2000; Manighetti et al., 2001, 2005; Hetzel et al., 2004).

When adjacent faults begin to overlap their stress fields interact, causing their displacement profiles to temporarily deviate from that which is predicted for a singular fault (Fig. 2. Once mechanical linkage of overlapping faults is accomplished, the composite fault will begin to recover unrealized displacement, and the displacement profile will return to roughly the same pattern as predicted for a single isolated fault (Fig. 2; Cartwright et al., 1995; Cowie and Roberts, 2001).

Researchers have exploited these evolutionary characteristics of normal fault systems to assess the geomorphic

response to step changes in the rate of local rock uplift (Boulton and Whittaker, 2009; Whittaker and Boulton, 2012; Whittaker and Walker, 2015). Following these previous studies and based on the observation that the Ptolemy and South-Central Crete faults linked in the geologically recent past (< 1 Ma), we infer that along the Tsoutsouros section of the fault uplift rates increased rapidly following linkage. Support for this inference comes from the mismatch in the mid-to-late Quaternary displacement pattern from marine terraces and the long-term displacement pattern observed in the present-day

topography (Fig. 1). We use this site of geologically recent fault linkage to explore the landscape response to a step change in tectonic forcing and to test the ability of the stream power incision model to predict the response of rivers to tectonic perturbations.

**2.5 River profile analysis**

The steepness of bedrock river channels can be used to infer rock uplift or erosion rates across a landscape.

Functional relationships between channel steepness and erosion rate have been empirically defined for a variety of geological settings where sufficient data exist (Safran et al., 2005; Harkins et al., 2007; Ouimet et al., 2009; DiBiase et al., 2010; Miller et al., 2013; Harel et al., 2016). A theoretical explanation of the relationship between river channel steepness and erosion rate can be described using what is commonly referred to as the stream power incision model (Howard and Kerby, 1983; Howard, 1994).

Detachment-limited river incision, $E$, into bedrock is typically modeled as a function of upstream contributing drainage area and channel gradient and, when combined with mass-conservation, can be expressed as,

$$\frac{dz}{dt} = U - E = U - KA^m S^n \qquad (1)$$

where $dz/dt$ is the change in elevation of the channel bed with respect to time, $U$ is rock uplift rate relative to a fixed base level, $A$ is the upstream drainage area, $S$ is local channel slope, $K$ is a dimensional coefficient that incorporates variables

dependent on incision process, substrate, climate and hydrology of erosion (e.g. Whipple, 2004), and $m$ and $n$ are positive constants that depend on basin hydrology, channel geometry, and erosion processes (Howard et al., 1994; Whipple and



Tucker, 1999; Whipple et al., 2000). Under steady-state conditions, where incision rate and rock uplift rate are equal, $dz/dt = 0$, Eq. (1) can be solved for the equilibrium channel slope at a given drainage area:

$$S = (U/K)^{1/n} A^{-(m/n)} \qquad (2)$$

Equation 2 predicts a power-law relationship between channel gradient and drainage area that is often observed in landscapes and has the same form as the empirically derived scaling relationship referred to as Flint's law (Flint, 1974):

$$S = k_s A^{-\theta} \qquad (3)$$

Where $k_s$ describes channel steepness and $\theta$ represents the concavity of the river channel, which are referred to as the channel steepness index and the channel concavity index, respectively (Snyder et al., 2000). The benefit of Eq (3) is that the stream profile parameters of $k_s$ and $\theta$ can be readily estimated by a linear regression of $log\ S$ and $log\ A$ (Kirby and Whipple, 2012).

The form of Eq (2) is comparable to that of Eq (3) such that:

$$k_s = (U/K)^{1/n} \qquad (4)$$

$$\theta = m/n \qquad (5)$$

Assuming steady state conditions, erosion, $E$, can be substituted for the uplift term, $U$, in Eq (4), thus defining the relationship between the channel steepness index and erosion rate (Kirby and Whipple, 2001).

Theory and empirical observations show that $\theta$ is insensitive to changes in rock uplift rate or erosion rate, while $k_s$ is sensitive to such changes (Whipple and Tucker, 1999; Snyder et al., 2000; Kirby and Whipple, 2012). Nonetheless, $k_s$ will covary with $\theta$ making it difficult to compare channel steepness indices from drainage basins with different concavities. To avoid this complication and in order to attempt to isolate only the influence of tectonics and rock erodibility on river profile morphology a reference concavity, $\theta_{ref}$, is often used to remove the effects of concavity on the steepness index by deriving a normalized steepness index, $k_{sn}$ (Wobus et al., 2006; Kirby and Whipple, 2012). From theory $\theta$ should vary between ~0.3 to 0.7, and many empirical studies find that $\theta$ is around 0.45 for most river profiles near equilibrium or graded conditions (Snyder et al., 2000; Wobus et al., 2006; Kirby and Whipple, 2012). Researchers generally use a reference concavity of ~0.45 to calculate the normalized steepness index as a baseline for comparison for $k_{sn}$ among different studies; alternatively, a reference concavity for a study area can be empirically determined as the regional mean concavity observed from relict (upstream from slope-break knickpoints) channel reaches.

The link between stream-power theory and Flint's law is useful because $k_s$ (or $k_{sn}$) and $\theta$ can be estimated by logarithmic regression of local channel slope versus upstream contributing drainage area from data easily extracted from digital elevation models (Wobus et al., 2006). However, this approach introduces considerable noise into the data by taking the derivative of elevation with respect to channel distance, and thus making it sometimes difficult to isolate more subtle





signals of interest. In an effort to reduce noise in river profile analysis Perron and Royden (2013) introduced the integral approach, which is more commonly called $\chi$-analysis, which relies on a transformation of the horizontal coordinate for a river profile from distance to $\chi$, where $\chi$ is an integral quantity with units of length. Separating variables in Eq (2), assuming $U$ and $K$ are spatially invariant, and integrating yields:

$$z(x) = z(x_b) + \left(\frac{U}{K}\right)^{\frac{1}{n}} \int_{x_b}^{x} \frac{dx}{A(x)^{m/n}}$$
(6)

Where z is elevation and $x_b$ is base level. The trailing term on the right-hand side of the equation is unitless. Therefore, a reference drainage area $A_o$ is introduced such that:

$$z(x) = z(x_b) + \left(\frac{U}{KA_o^m}\right)^{\frac{1}{n}} \chi$$
(7)

Where:

$$\chi = \int_{x_b}^{x} \left(\frac{A_o}{A(x)}\right)^{m/n}$$
(8)

Equation 7 is convenient because it has the form of a line where $z$ is the dependent variable, $\chi$ is the independent variable, $z(x_b)$ is the y-intercept and $\left(\frac{U}{KA_o^m}\right)^{\frac{1}{n}}$ is the slope. Plots of $\chi$ and $z$ are referred it as $\chi$-plots. It is important to recognize that when $A_o$ is assumed to be 1 the slope in a $\chi$-plot is the same as $k_{sn}$ from slope-area analysis. Alternatively, if $A_o$ is a value other than unity, simply multiplying the slope of a $\chi$-plot by $A_o^{-m/n}$ will give $k_{sn}$. Because $A_o$ is an arbitrarily
selected value, we suggest always using a value of 1 for $A_o$ in an attempt to standardized $\chi$-plots and ensure that the slope of the $\chi$-plot can be directly interpreted in terms of the commonly used channel metric $k_{sn}$.

Based on Eq. (2), a river will respond to a change in rock uplift rate by steepening its gradient, while its concavity will remain largely unaffected (Snyder et al., 2000). Changes in channel gradient in response to a temporal change in rock uplift will initiate at the river channel mouth and progressively propagate headword until the river gradient has equilibrated
to the new uplift regime (Fig. 2c; Whipple and Tucker, 1999). The signal of a change in the rate of uplift is transmitted upstream through the propagation of a knickpoint, a convex-upward reach in an otherwise concave-upward river longitudinal profile (Fig. 2c; Whipple and Tucker, 1999). The knickpoint represents the mobile boundary between the "relict" portion of the landscape unaware of a change in rock uplift rate above the knickpoint and the part of the landscape that is adjusted or adjusting to the newly imposed uplift rate below the knickpoint. In such cases, knickpoints are identified as breaks in the
logarithmic regression of local channel slope versus contributing drainage area, or a break in slope between two linear segments in a $\chi$-plot (Fig. 2c). The channel steepness indices for the river reaches above and below the knickpoint represent the relative change in the rate of uplift. Thus if fluvial knickpoints are present, temporal changes in the rate of rock uplift can be inferred from analysis of river profiles (e.g. Miller et al., 2012, 2013). By determining the normalized steepness index,





river channels can be used to infer and map the patterns of relative rock uplift rate in a given region, provided that lithologic contrasts and spatial variations in climate are marginal in relation to variations in rock uplift (Kirby and Whipple, 2001).

## 2.6 Drainage basin stability analysis

The transformed distance variable $\chi$, has been shown to be a useful metric for interpreting the 'dynamics' or relative stability of catchment divides when plotted in map view (Willett et al., 2014). Analysis of $\chi$-maps relies on the assumption that nearby channel heads in adjacent drainage basins will have the same $\chi$ value at equilibrium. Provided that nearby channel head elevations are roughly the same, which is a reasonable assumption, $\chi$ at a channel head represents a measure of the aggregate steepness of that channel. Lower $\chi$ at a channel relative to neighboring channel heads indicates a steeper and more rapidly eroding channel, whereas a higher $\chi$ value at a channel head relative to heads of neighboring streams suggests a gentler and more slowly eroding channel. All else being equal (e.g. uplift rate, bedrock erodibility, precipitation, etc.), local drainage divides should migrate in the direction of high $\chi$ simply because those basins are eroding more slowly than neighboring basins (Willett et al., 2014). By producing a map of river network $\chi$ the dynamics or stability of a drainage network is illuminated by simply quantifying contrasting $\chi$ values across local drainage divides.

## 3 Methods

### 3.1 Coastal uplift function

Derivation of coastal uplift rates is from data reported in Gallen et al. (2014) who used optically stimulated luminescence dating of the lowest elevation marine terraces to anchor the sequences to the mid-to-late Quaternary sea level curve. Based on stratigraphic and sedimentological evidence demonstrating that marine terraces in Crete are cut and deposited during transgression to sea level high stands and abandoned during sea level regression, Gallen et al. (2014) estimated the age of successively higher, older terraces by tying them to consecutively older sea level high stands (e.g. Merritts and Vincent, 1989).

Mid-to-late Quaternary uplift rates derived from dated marine terrace sequences were used to model uplift rates as a function of distance along the coast (Fig. 3). First, site-specific average uplift rates were determined using all terraces, geochronologic age constraints, and assigned ages for each site-specific flight of terraces reported in Gallen et al. (2014) (Fig. 3a). To determine Quaternary site average uplift rates, a Monte Carlo sampling approach was adopted to incorporate uncertainties from all measurements, and for each site five-thousand Monte Carlo-regression simulations were carried out. For each simulation random samples were drawn from a normal distribution based on the mean and 1-σ uncertainty of each measurement of terrace elevation, geochronologic age, and eustatic elevation at the time of terrace formation and used to calculate the total change in the elevation of each individual terrace. A linear regression through the theoretical terrace age versus total elevation change data was used to determine the average uplift rate (Fig 3a-inset). After the simulations were



complete the mean and 2-σ uncertainties of the five-thousand theoretical uplift rate calculations were determined and used as the average uplift rate for each site.

Second, following a similar approach, Monte-Carlo simulations using five-thousand samples from the site average uplift rates along the Asterousia footwall and the Dikti hanging wall blocks of the SCCF were used to generate a model of Quaternary uplift rate as a function of distance along the coastline (Fig. 3b). For each fault block, a second-order polynomial regressions was used because it best approximates the observed pattern of uplift from the mapped terraces (Fig. 3b). The model of coastal Quaternary uplift rate was used to estimate the uplift rate at the outlets of the 21 catchments used in this study and was compared to the normalized channel steepness ($k_{sn}$) for lower segments of basin trunk channels.

### 3.2 Topographic analyses

A commercially available NEXTmap® world 30$^{tm}$ v2.0 30-m resolution digital elevation model (DEM) was used to analyze the topography of south-central Crete. The World 30 DEM is a preprocessed product that aggregates SRTM3 v2.1, ASTER v2.0 and GTOPO30 DEMs to improve the vertical accuracy, fill data voids, and reduce noise. All topographic analyses were conducted using the MATLAB software package TopoToolbox Version 2 (Schwanghart and Kuhn, 2010; Schwanghart and Scherler, 2014). Local slope was calculated from the filled DEM using a D8 algorithm (Fig. 4). Only basins draining ≥ 3 km$^2$ were analyzed because it has been shown that ephemeral streams, such as those on Crete, have difficulty responding to tectonic forcing at small drainage areas (e.g. Frankel and Pazzaglia, 2005). Twenty-one drainage basins were delineated and are labeled 1 to 21 from west to east (Fig. 4). There are 11 drainage basins in the Asterousia (1-11) and 10 in the Dikti (12-21). Basin size ranges from 4 km$^2$ to 96 km$^2$ with a mean and standard deviation of $18 \pm 15$ km$^2$. Trunk channels were extracted from each of the 21 basins and used for river profile analysis. The initiation points of the trunk channel for each drainage basin in the study was set at a basin area of 0.1 km$^2$, where we observed a break in local channel slope versus contributing drainage area. This scaling break is often interpreted as the along-channel transition from hillslope and landslide dominated erosional processes to the portion of the channel where erosion occurs primarily by fluvial processes (e.g. Montgomery and Foufoula-Georgiou, 1993; Wobus et al., 2006).

We used 1:50,000-scale geologic maps to evaluate the role of lithology on fluvial and drainage basin metrics (Fig. 1c; Greek Institute of Geology and Mineral Exploration, 1972; 1974; 1977; 1987). The geologic maps were digitized and rock types classified as Mesozoic mudstones and sandstones, Mesozoic carbonates, Neogene turbidites, or other, which comprises Mesozoic ophiolites, and Cenozoic phyillite-quartzite nappes, and Quaternary deposits that collectively comprise < 10 % of the study area (Figs. 1c). The geologic map was used to determine the rock types that each river flows over to visually assess for lithologic controls on river profile geometry.



### 3.3 River profile analysis

Trunk channels of the 21 river basins were extracted using TopoToolbox version 2 (Schwanghart and Kuhn, 2010; Schwanghart and Scherler, 2014) and a series of MATLAB functions for analyzing river profiles that are available from the corresponding author upon request. In our analysis, sinks in the raw DEM were filled and the resulting filled DEM was used to generate flow direction and flow accumulation grids. Trunk channels from the 21 river basins were extracted using a steepest-decent algorithm. Data extracted from the filled DEM and hydrologically modeled grids include elevation, distance from river mouth (in this case sea level), distance from channel head, and catchment area. The elevation data was smoothed using a 500 m moving window average to reduce noise in the river profiles.

Chi was calculated by integrating drainage area with respect to channel distance from sea level (base level) using Eq. (8) and an $A_o = 1$. Maps of $k_{sn}$ for the entire stream network and for individual trunk-channel segments were generated by preforming a linear regression through $\chi$ and the smoothed channel elevation data along 500 m channel segments. This approach is similar to that outlined by Wobus et al. (2006) who used power law regressions through slope verse drainage area data for river segments of a given size; however, the approach described here requires far less smoothing of the data, and is thus a more computationally efficient method for $k_{sn}$ map generation.

Linear regressions through $\chi$-elevation data were performed by manually selecting upper and lower bounds along individual channels. The 95 % confidence intervals for reported $k_{sn}$ values are based on the uncertainties in slope of the regression due to scatter in the $\chi$-elevation data. Channel segments were defined on the presence of knickpoints (breaks in channel slope in $\chi$-plots). Knickpoints were defined at locations where $k_{sn}$ decreased 25% relative to the next downstream channel segment. We also defined some lower segments of river profiles based upon where known active faults cross the channels.

### 4 Results

### 4.1 Site average uplift rates and coastal uplift model

The site average uplift rates calculated for the Ptolemy footwall (Asterousia Mtns) range from 0.4 to 0.8 m kyr$^{-1}$ and generally increase from west to east (Fig. 3b). The 2$^{nd}$-order polynomial fit to the site average uplift rates is significant at 95% confidence and describes the data well with an r$^2$ of 0.74. Site average uplift rates in the hanging wall of the South-Central Crete fault (Dikti Mtns) are between 0.15 and 0.35 m kyr$^{-1}$ with the lowest values found ~15 to 20 km from the western end of the fault block. The 2$^{nd}$-order polynomial fit to the data is also significant at 95% confidence, but does not describe the data quite as well as the fit for the Ptolemy footwall with an r$^2$ of 0.56 (Fig. 3b). Nonetheless, the fit is reasonably good and follows a broad synclinal-style warping (Fig. 3b).



## 4.2 Basic topographic and fluvial metrics

Topography is generally steep at low elevations along the coast of the Asterousia Mountains and a major break in slope is noted ~ 2 – 3 km inland at elevations between 0.2 – 1 km (Fig. 4a). Topographic slope at high elevation in the Asterousia is typically < 20°, with the exception of some high elevation peaks (Fig. 4a). Steep topography in the Dikti footwall is observed at low elevations just to the north of the South-Central Crete fault and around high elevation peaks (Fig. 4a). Intermediate elevation low gradient topography is noted in many of the study catchments, such as catchment 13, 16, and 19. Slope gradients are generally low in the hanging wall of the South-Central Crete Fault (Fig. 4a).

Maps of normalized channel steepness index show patterns that mimic the distribution of local channel slopes (Fig. 4). The range front of the Asterousia has steep channels, many with $k_{sn}$ values > 200 m$^{0.9}$ (Fig. 4b). Channel segments at higher elevations between ~0.5 – 1 km elevation are low-gradient with $k_{sn}$ values generally < 70 m$^{0.9}$ (Fig. 4b). In the Dikti there is an abrupt change in $k_{sn}$ across the South-Central Crete fault; $k_{sn}$ values in the hanging wall downstream of the fault are mostly < 50 m$^{0.9}$ and locally increase to > 100 m$^{0.9}$ upstream of the fault (Fig. 4b). Rivers draining the high peaks of the Dikti have $k_{sn}$ values between 100 – 150 m$^{0.9}$. Along the footwall of the South-Central Crete fault between the fault and the main drainage divide to the north, local river reaches have $k_{sn}$ values that decrease to below 60 m$^{0.9}$ (Fig. 4b).

## 4.3 Trunk channel knickpoints, longitudinal profiles, and the channel steepness index ($k_{sn}$)

Trunk channel rivers in basins 1 – 9 and 12 in the Asterousia Mountains only contain a single knickpoint while trunk rivers in basins 10, 11, and 13 – 21 all contain two. With the exception of basin 12, those rivers that contain two knickpoints directly cross over the South-central Crete fault (Fig. 5a). Most knickpoints do not correlate with a specific mapped geologic contact and knickpoints are found in a variety of rock types, suggesting rock type isn't the dominant factor controlling knickpoint formation (Fig. 5b). All knickpoints are of the slope-break variety, indicating that their channels exhibit a sustained change in channel steepness above and below the knickpoints (Fig. 5c-e; Kirby and Whipple, 2012).

Knickpoints are classified as generation-1 or generation-2 based on landscape position, topographic characteristics, and relative changes in $k_{sn}$ above and below the knickpoints (Fig. 5b). Generation-1 knickpoints are at higher elevations, relative to generation-2, and are found in all rivers except for basin 12 (Fig. 5b). Elevations for generation-1 knickpoints range from several hundred meters to > 1.5 km (Fig. 5b). Generation-2 knickpoint elevations are highest in the center of the Asterousia and Dikti footwalls and progressively decline in elevation away from the center of each fault block (Fig. 5b). Generation-2 knickpoints are only found along rivers that drain directly over the South-Central Crete fault (Basins 10 – 21) (Fig. 5a, b). These knickpoints are < ~ 0.5 km in elevation and most are found < 5 km from the river mouth (Fig. 5b). The knickpoint elevations are scattered but are generally higher near the eastern and western ends of the South-Central Crete fault (Fig. 5b).



The river longitudinal profiles are generally smooth and concave upward between convex-upward reaches (knickpoints) (Fig. 5b). Rock type is correlated with high frequency noise in the river profiles, but does not appear to have first-order control on river profile morphology. Most trunk channel segments between knickpoints are near linear on $\chi$-elevation plots (Fig. 5c-e). This observation indicates that the reference concavity of 0.45 is a reasonable approximation of

the concavity of south-coast Cretan rivers (e.g., Perron and Royden, 2013; Mudd et al., 2014). Some river profiles contain minor knickpoints that can be related to local changes in rock type, such as reach 2 along stream 10 (Fig. 5d). Some streams (13, 16, and 18) have low gradient reaches where they are incising fluvial and lacustrine deposits.

In figure 6 we plot $k_{sn}$ versus west-to-east distance along the coast for different river segments. From base level, the lowest segment for each stream is the Ptolemy footwall in the Asterousia Mountains and the hanging wall of the SSCF

along the Dikti range front (Fig. 6a). The lowest segment $k_{sn}$ values are higher in the Asterousia footwall, with a mean of 220.8 ± 107.5 m$^{0.9}$ (1$\sigma$), when compared to a mean $k_{sn}$ of 51.8 ± 27.4 m$^{0.9}$ measured from streams flowing across the SCCF hanging wall in front of the Dikti Mountains. Along the Ptolemy fault footwall, $k_{sn}$ is low in the west and progressively increases to about kilometer 20 then declines slightly approaching the onshore exposure of the South-Central Crete fault (Fig. 6a). The pattern of $k_{sn}$ along the hanging wall segment of the Dikti range is consistently low, but minor increases in k$_{sn}$

are noted at the western and eastern ends of the fault block (Fig. 6a). k$_{sn}$ values in the lowest reach of the Ptolemy and SCCF footwalls are similar, with a mean of 267.0 ± 96.5 m$^{0.9}$ from the Dikti range (Fig. 6b). There are 12 rivers that contain two knickpoints (basins 10 – 21) and the reach between these knickpoints has a mean $k_{sn}$ of 117.5 ± 53.3 m$^{0.9}$, which is consistently lower than the next downstream segment (Fig. 6c). As a function of distance along the coastline, $k_{sn}$ values for the middle footwall reach are low where the SCCF transitions offshore to the Ptolemy fault near Tsoutsouros (Fig. 6,

kilometer 35 – 40). East of Tsoutsouros $k_{sn}$ values progressively increases to about kilometer 55, then decline toward the eastern end of the SCCF block (Fig. 6c). The $k_{sn}$ values for the upper footwall reaches of both mountain ranges are consistently low with a mean of 52.2 ± 24.3 m$^{0.9}$ (49.4 ± 21.7 m$^{0.9}$ in the Asterousia and 55.7 ± 26.6 m$^{0.9}$ in the Dikti, respectively) (Fig. 6d). No obvious along the coast pattern is observed from the Asterousia range front, but along the Dikti range $k_{sn}$ is slightly elevated towards the middle of the footwall, with lower values at either ends of the fault block (Fig. 6d).

The lowest stream reaches along the Ptolemy footwall (Asterousia range front) and SCCF hanging wall (Dikti range front) should be the most sensitive to uplift, as is recorded by the Pleistocene marine terraces. We observe a monotonic increase in $k_{sn}$ as a function of average modeled Quaternary uplift rate for the channel reaches that discharge to base level, but the relationship does not appear linear (Fig. 7). We fit the data with a power-law function, an unconstrained linear regression and a linear regression forced through the origin, all conducted using an orthogonal least-squares method. The

data is best fit by a power-law function with an exponent of 0.55 (Fig. 7a). This regression is significant at > 99% confidence with an r$^2$ of 0.71. An unconstrained linear fit to the data is also significant at > 99% confidence and has an r$^2$ of 0.61 (Fig. 7b) a linear it forced though the origin is significant at ~ 95% confidences but poorly describes with data with an r$^2$ of 0.4 (Fig. 7c). Channels are steeper when underlain by Mesozoic carbonates and mud-and-sandstones relative to channels



underlain by Neogene turbidites and other rock units (Fig. 7). However, these differences coincide with a large change in uplift rate; the Mesozoic units are in the more rapidly uplifting Ptolemy footwall of the Asterousia range front, while the Neogene and other rock units are confined to the more slowly uplifting hanging wall of the SCCF along the Dikti range front (Fig. 6a, 7).

**5 Discussion**

**5.1 Long-term versus intermediate vertical displacement patterns**

The pattern of displacement and uplift rates recorded by the marine terraces is discordant with the long-term record of fault displacement as recorded by topography (Fig. 1, 3). The maximum envelope of topography is consistent with displacement along two independent fault systems; the offshore Ptolemy fault and the on-shore South-Central Crete fault

(Fig. 1). The Asterousia range exhibits a roughly parabolic displacement profile approximated by maximum range elevations, and the displacement-to-fault length ratio is 0.034, consistent with expectations from fault mechanics theory (Fig. 1; Cowie and Scholz, 1992; Drawers et al., 1993; Schlishe et al., 1996). Similarly, the topographic envelope of the Dikti range is triangular with a maximum in the middle that tapers to minimum values at the fault tips (Fig. 1). The Dikti range has a displacement-to-length ratio of 0.032. A topographic minimum is observed where the South-Central Crete fault takes a jog

and extends offshore near the town of Tsoutsouros (Fig. 1). This topographic minimum is interpreted as a relict feature resulting from a long-term slip deficit (c.f. Cartwright et al., 1995) that has not yet been removed by slip accumulated on the relay between the two now-linked faults. We can roughly bracket the timing of fault linkage based on previous studies and observations. A minimum age of ~400 kyrs is obtained from the inferred age of the oldest marine terrace cut by the fault (Fig. 3a). A back-of-the-envelope maximum age for linkage of ~ 0.6 – 1.0 Ma is determined by dividing the modern height

of the footwall above the fault trace in the linkage zone (500 – 600 m) by the footwall uplift rate in the linkage zone (0.6 – 0.8 mm yr$^{-1}$).

The pattern of deformation recorded by marine terraces in south-central Crete supports the interpretation of a linked normal fault bounding the Asterousia and Dikti mountains with the most rapid uplift for at least the last 400 ka observed near the linkage zone (Fig. 1, 3). The expected pattern of uplift in the hanging wall of the South-Central Crete fault also appears

to support this interpretation, but it is not as straight forward as it is along the Asterousia range front (footwall of the Ptolemy fault) (Fig. 3). Assuming that the only contribution to vertical tectonics in the hanging wall of the SCCF were uniform regional uplift and slip on the South-Central Crete fault, we would expect slower uplift (or possibly subsidence) near the linkage site that increases to the east. Uplift rates near Tsoutsouros are lower than at the eastern end of the study area, but the slowest rates are found near the center of the South-Central Crete fault. Low uplift rates near the fault center are due to local

subsidiary, orthogonally-striking faults embedded in the hanging wall of the SCCF (Gallen et al., 2014). Another factor that may play a role is uplift rate determinations is proximity of the mapped marine terraces to the master fault, as displacements





will be greatest at the fault and decay with orthogonal distance from the fault. Terraces are closest to the South-Central Crete Fault near the fault center (near the town of Arvi) (Fig. 1b). Despite this discrepancy the site average uplift pattern and the pattern of marine terrace deformation are consistent with displacement on a linked Ptolemy Fault-South-Central Crete Fault system. The above is simply a defense of the interpretation that the Ptolemy fault and the South-Central Crete fault recently

linked, which provides the basis for this investigation.

**5.2 River profiles, knickpoints, and topography**

We interpret relative differences in slope and $k_{sn}$ as reflecting changes in landscape-scale erosion rates across the study area (Fig. 4). This interpretation is supported by important empirical studies that demonstrate positive relationships between topographic (e.g., slope) and fluvial (e.g. $k_{sn}$) metrics (Ahnert, 1970; Montgomery and Brandon, 2002; Ouimet et

al., 2009; DiBiase et al., 2010; Kirby and Whipple, 2012; Miller et al., 2013). Further, the stream power incision model predicts that, all else being equal, a steep-gradient river channel will incise more rapidly than will a shallow-gradient channel (Howard et al., 1994; Whipple and Tucker, 1999). Within this framework knickpoints represent the mobile boundary separating river reaches that are eroding at different rates (Whipple and Tucker, 1999; Kirby and Whipple, 2012).

The steep hillslopes and high $k_{sn}$ along the range front of the Asterousia Mountains are interpreted as reflecting a

recent phase of increased erosion (Fig. 4a, b). In the nine westernmost basins (basins 1 – 9) there is a break in topographic slope and $k_{sn}$ between 300 and 1000 m above modern sea level that is controlled by a single fluvial knickpoint on the channel of each basin (Figs. 1, 4, 5). This marked landscape transition suggests a reduction in the rate of erosion at higher elevations that we interpret to be largely the result of an increase in uplift rate of the Asterousia footwall. In the eastern two catchments in the Asterousia (basins 10 and 11) two knickpoints are observed. The $k_{sn}$ values of the three river reaches

bound by these knickpoints progressively increases upstream (Figs. 4b, 5d). We interpret this distribution of $k_{sn}$ as the result of a two phase increase in uplift rate, where each jump in uplift rate generated a knickpoint.

The changes in topographic slope and $k_{sn}$ are noisier for Dikti streams when compared to those draining the Asterousia. This difference is mostly due to greater variation in local rock type for Dikti-draining streams relative to the Asterousia (Fig. 1a, 5b). Despite the noisier signal, general patterns are observed from the Dikti river profiles that are similar

to those noted in the Asterousia (Figs. 4, 5). Topographic slopes and $k_{sn}$ are low along the SCCF hanging wall of the Dikti Mountains, likely reflecting low erosion and uplift rates (Fig. 4, 5). Across the South-Central Crete fault there is an abrupt change in both local slope and $k_{sn}$ indicating an abrupt change in the rates of both rock uplift and erosion from the hanging wall onto the footwall sides of the fault (Fig. 4b, 5e). River profiles in the SCCF footwall are morphologically similar to basins 10 and 11 in the Asterousia (Ptolemy fault footwall), with all profiles exhibiting two major knickpoints where $k_{sn}$

progressively decreases upstream (Fig. 4-5d, e). Similar to the eastern end of the Asterousia, we interpret the formation of knickpoints and progressive steepening of the river profiles as the result of a two phased increase in uplift rate.



Generation-1 knickpoints are interpreted to record an early increase in the uplift rate and generation-2 knickpoints the tell-tale signal of a more recent increase. The river reaches above generation-1 knickpoints are uniformly shallow suggesting a period of relatively slow uplift rate (Fig. 5d-e, 6). The elevation pattern of generation-1 knickpoints is consistent with a cumulative uplift history on two individual normal fault segments with the highest elevations at fault center in each footwall that taper to lower elevations at the fault tips (Fig. 5b). We thus interpret generation-1 knickpoints as being created by an increase in footwall uplift rate due to accelerated slip along independent Ptolemy and South-Central Crete faults prior to fault linkage. Because of their spatial proximity and association with the South-Central Crete fault, we infer that the lower elevation generation-2 knickpoints record a change in footwall uplift rate associated with a change in rate of fault slip due to linkage of the South-Central Crete fault and Ptolemy faults (Fig. 5). To further explore this hypothesis of river profile morphology and knickpoint generation we turn to the $k_{sn}$ pattern preserved in individual longitudinal profile segments as a function of distance along the coast as a means to uncover the relative uplift history in space and time and compare it to expectations from fault mechanics (Fig. 2, 6).

**5.3 Normalized Steepness index ($k_{sn}$) patterns and uplift history**

**5.3.1 Along strike patterns**

The interpretations presented in the following section rely on the stream power incision model described in section 2.5. Specifically we assume that $k_{sn}$ largely reflects the ratio of uplift rate to bedrock erodibility per Eq. (4). We emphasize the phrase 'largely reflects' because, as argued in section 5.5, there are factors other than uplift rate and erodibility (e.g. waterfall formation, temporal variations in catchment area) that can influence the empirically derived value of $k_{sn}$. Moreover, we acknowledge that rivers do not only respond to changes in uplift through river channel steepening, as river channel width and the size of grains delivered to channel segments also evolve in space and time during transient landscape evolution (Turowski et al., 2006, 2009; Amos and Burbank, 2007; Whittaker et al., 2007; Finnegan et al., 2007). Despite these complexities, we argue that the dominant factor influencing the development of knickpoints and changes in river profile $k_{sn}$ from the streams investigated in South-central Crete are variations in the relative rate of rock uplift.

The existence of dated marine terraces preserved in the Asterousia footwall and the Dikti hanging wall allows for comparisons of both $k_{sn}$ and uplift rates for the lowest river reaches (Fig. 3b, 6a). Along the Ptolemy footwall of the Asterousia Mountains, $k_{sn}$ from the lowest river reaches are on average four times higher than $k_{sn}$ from comparable reaches of the SCCF hanging wall in front of the Dikti Mountains. Non-coincidentally, this difference in $k_{sn}$ mimics the differences in rock uplift rate between the two fault blocks, which differ by a factor $> 2.5$ (Fig. 3b, 6a). The larger difference in $k_{sn}$ between the two fault blocks when compared to uplift rate might partially reflect differences in bedrock erodibility between river reaches along the Ptolemy fault footwall, which drain mostly more resistant Mesozoic carbonates and sandstones and mudstones, versus those draining the SCCF hanging wall, which cut through less indurated Neogene turbidities. However, for reasons outlined in the following section, 5.3.2, we think that much of this difference has to do with a non-linear



relationship between channel erosion and uplift, where incision becomes less efficient at higher uplift rates. Along the Asterousia range front, $k_{sn}$ of the lowest reaches increases from west-to-east approaching the middle of the fault and remains steady or slightly declines approaching the western tip of the South-Central Crete fault (Fig. 6a). This pattern mimics the displacement field recorded by marine terraces (Fig. 1d, 3b). The relatively low values of $k_{sn}$ in the hanging wall of the

SCCF are consistent with the low uplift rates recorded by marine terraces along this portion of this fault block. The monotonically-increasing relationship between $k_{sn}$ for the lowest river reaches and the modeled coastal uplift rate supports the interpretation that $k_{sn}$ is sensitive to changes in rock uplift rate (Fig. 7).

The lowest channel reaches along the footwalls of both the Asterousia and Dikti Mountains show similarly high $k_{sn}$ (Fig. 6b). When plotted along the coastline this pattern is interpreted to reflect the present-day relative uplift rate of the

footwalls, suggesting that the lowest uplift rates are located at the western and eastern tips of the Ptolemy and South-Central Crete faults, respectively. Both values of $k_{sn}$ and uplift rate progressively increase approaching the center of the study area (town of Tsoutsouros) for ~ 20 km in either direction (Fig. 6b). This pattern is consistent with the displacement and footwall uplift pattern expected for two linked normal faults, with the most rapid uplift rates near the zone of linkage that then decline to minimum values approaching the outer tips of each fault (Fig. 2, 6b). While a side note, it is important to recognize that

because the SCCF is onshore along the Dikti range front, it is possible to observe changes in $k_{sn}$ as streams cross from the hanging wall onto the footwall. In all instances, $k_{sn}$ values increase across this structural transition, and as such indicate that the fault is active (Fig. 6a, b).

The middle reaches of basins 10 – 21 exhibit lower $k_{sn}$ relative to the lowest footwall reach, indicating a relatively slower uplift rate. We interpret the absence of a lower knickpoint in basins 1 – 9 as an indication that the uplift rate west of

basin 9 did not change significantly after linkage of the Ptolemy and South-Central Crete faults. Assuming this interpretation is correct, the relative uplift rate for the lowest reach in basins 1 – 9 allows one to interpret the pattern of uplift along the Ptolemy and South-Central Crete faults prior to linkage (Fig. 6c). The pattern of $k_{sn}$ for footwall-draining streams is consistent with the uplift pattern expected on two mechanically isolated faults, with the highest values at the fault center that then taper to minima at the fault tips (Fig. 2, 6). The along strike pattern of $k_{sn}$ for these stream reaches is consistent with the

pattern of total cumulative displacement recorded by the maximum envelope of topography and the elevation distribution of generation-1 knickpoints (Fig. 1c, 5b, 6c).

The upper most river profile segments, those above generation-2 knickpoints, show consistently low $k_{sn}$ indicating relatively low uplift rate over the entire study area (Fig. 6d). Our interpretation is that many of these low-gradient reaches record pre-fault topography and the background regional rates of rock uplift. Following from this interpretation, we consider

many of the upland portions of the river basin as "relict topography"; however, we later argue in section 5.5 that not all of this topography is relict in the traditional view of a static drainage network geometry through time (c.f., Gallen et al., 2013), as portions of a few of the catchments may record geologically recent integration of basins that were internally drained.



In summary, the patterns of $k_{sn}$ for different river reaches reflect a three phase uplift history. Starting with the oldest and moving forward in time, the upper most river reaches above generation-2 knickpoints are interpreted as recording a period of regionally low uplift rate (Fig. 6d). Intermediate river reaches follow an uplift pattern consistent with the pattern of displacement along two independent normal fault systems with a maximum uplift rate near the fault centers that then

tapers to minima at the fault tips (Fig. 6c). Normalized channel steepness values from the lower footwall reaches are consistent with footwall uplift following linkage of adjacent fault systems, as increased uplift rates occur near the site of fault linkage (Fig. 6b), while Ksn values from the hanging wall of the South-Central Crete fault are consistent with lower rates of net (secular + local) rock uplift due to slip on the fault (Fig. 6a). This history is consistent with the conceptual model presented in figure 2 and supports the findings of numerous other studies that show that river profiles are sensitive recorders

of relative uplift histories (Snyder et al., 2000; Kirby and Whipple, 2001, 2012; Wobus et al., 2006; Whittaker et al., 2008; Boulton and Whittaker, 2009; Pritchard et al., 2009; Whittaker and Boulton, 2012; Perron and Royden, 2013; Royden and Perron, 2013; Goren et al., 2014; Whittaker and Walker, 2015).

**5.3.2 Relationship between uplift rate and $k_{sn}$**

The modeled coastal uplift pattern derived from marine terraces is similar to the relative pattern of $k_{sn}$ along the

coastline for the lowest river reaches along the Ptolemy footwall of the Asterousia range and river reaches on the hanging wall of SCCF in front of the Dikti Mountains (Fig. 7). The empirical estimation of stream power parameters $K$, the erodibility constant, and $n$, the slope exponent is achieved by assuming that these river reaches are locally in equilibrium, and by rearrangement of Eq. (4) (Fig. 7). We again acknowledge that in transient landscapes rivers not only adjust to changes in rock uplift rate through river channel steepening, but also by reduction in channel width in concert with spatial

and temporal changes in sediment flux and grain size (Sklar and Dietrich, 2004; Finnegan et al., 2005, 2007; Amos and Burbank, 2007; Whittaker et al., 2007; Cook et al., 2013; Attal et al., 2015; Shobe et al., 2016). Nonetheless, we suggest that the rivers in south-central Crete predominantly steepen in response to increased uplift rates and therefore regression through $k_{sn}$ – uplift rate data by an orthogonal least-squares method can be used to empirically calibrate parameters in the stream power incision model under different assumptions.

We first assume that the relationship between $k_{sn}$ and uplift rate is non-linear and fit the data with a power-law regression that yields a regionally averaged erodibility constant, $K$, of ~ 3.47e$^{-5}$ m$^{0.51}$ yr$^{-1}$ and defines the slope exponent, $n$, as 0.55 (Fig. 7a). Assuming that n = 1 and fitting the data with a linear regression we derive a $K$ of ~ 1.58e$^{-6}$ m$^{0.1}$ yr$^{-1}$ when the fit is unconstrained (Fig. 7b) and a $K$ of ~ 2.69e$^{-6}$ m$^{0.1}$ yr$^{-1}$ when the regression is forced through the origin (Fig. 7c). Both the power-law and unconstrained linear regression described the data well with r$^2$ of 0.71 and 0.61, respectively (Fig. 7a and

b). However, with an r$^2$ of 0.05, the linear regression forced through the origin provides a poor fit to the data (Fig. 7c). All of the regionally average $K$ values are consistent with the range of other studies (e.g., Stock and Montgomery, 1999); however, the only fit that realistically describes the data is the power-law fit (Fig. 7). Our rational is that the linear fit forced though





the origin does not describe the data well and the unconstrained linear fit has a y-intercept on the uplift rate (a proxy for local erosion rate) axis. This would suggest that there is a threshold uplift rate (or erosion rate) at which the rivers begin to steepen. Prior to this threshold the rivers would be capable of incising, but with zero steepness, which is unreasonable. Rather, most studies that empirically calibrate the stream power incision model find a y-intercept on the $k_{sn}$ axis if a linear model were used suggesting there is a threshold channel steepness that must be attained before erosion commences, which is more physically reasonable (e.g. Miller et al., 2013). We therefore interpret the power-law fit as the best and most physically reasonable approximation of our data.

An interesting outcome of this analysis is that the slope exponent for rivers in south-central Crete is ~ 0.55. Most studies that empirically calibrate the exponents needed for the steam power equation derive values of $n$ that are $\geq 1$ (Ouimet et al., 2009; DiBiase et al., 2010; Whittaker and Boulton, 2012; Harel et al., 2016). A slope exponent, $n$, of less than one is interesting because it suggests that river incision becomes less efficient with steepening of the river channel. Rivers in the central Apennines that are in a tectonically, geologically and climatically similar environment to Crete, with transient landscapes, active extensional faults, significant carbonate bedrock and a semi-arid climate are also reported to have $n < 1$ (Whittaker et al., 2008; Attal et al., 2011; Royden and Perron, 2013; Mudd et al., 2014). We do not think that the similarities between the observations in Crete and the central Apennines are coincidental and suggest that processes that operate to produce $n < 1$ (discussed in the section 5.5) in south-central Crete might apply elsewhere.

**5.4 Knickpoint metrics**

If the rivers in south-central Crete behave in a manner consistent with the stream power incision model, knickpoint distributions should be predictable. Differences in knickpoint elevation will occur due to spatially variable rock uplift, which is observed as noted above. Models of knickpoint retreat based on the stream power incision model (e.g., Rosenblum et al., 1994; Whipple and Tucker, 1999; Crosby and Whipple, 2006; Berlin and Anderson, 2007) state that the celerity of a knickpoint will be:

$$\frac{dx}{dt} = KA^mS^{n-1} \tag{9}$$

It is important to note that Eq. (9) does not incorporate a component of uplift, such that knickpoint celerity is not dependent on uplift rate. This equation can be solved for $dt$ and, in the special case where $K$, $m$, and $n$ are known, integrated to determine the response time of the river, which is referred to as $\tau$ (Goren et al., 2014; Fox et al., 2014). The response time of the river from a reference base level position to a knickpoint represents the knickpoint travel time, and knickpoints originating from the same relative base level fall event should have the same $\tau$. Herein, we utilize the variable $\tau'$ to indicate to readers that the time component associated with $\tau$ determinations in our study is relative, not absolute, as it is based on empirically derived constants that may not represent actual values. The calculation of $\tau'$ is valuable in areas of spatially variable uplift rate, such as south-central Crete, because the elevation of knickpoints from the same relative base level fall



event will not lie at the same elevation, as is expected if uplift rate is uniform (Niemann et al., 2001). It is also worth noting that if $n = 1$ and $K$ is uniform than $\chi$ effectively represents the relative response time of a river network and, thus, knickpoints from a common base level fall event will cluster around the same $\chi$ distance.

The knickpoints in south-central Crete fail to conform to predictions of travel distance based on the linear form of Eq. (9), which predicts that knickpoints of a common origin will be at the same $\chi$ distance (Fig. 8a). Both generations of knickpoints have a range of $\chi$ distances and knickpoint $\chi$ distance has an apparent weak inverse correlation with downstream $k_{sn}$ (Fig. 8a, b). This latter observation is expected if $n < 1$, and therefore supports our analysis of $ksn$ and uplift rate data that suggests n ~ 0.55 (e.g. Fig. 7a). However, when the river response time, $\tau'$, is calculated using the empirically calibrated $K$ and $n$ from the power-law fit shown in figure 7a both knickpoint generations fail to collapse onto a single time (Fig. 8c).

We note here again that we do not interpret $\tau'$ calculated here as the actual response time of the rivers in south central Crete, but rather as an apparent response time   for reasons we discuss in the follow sections. Generation 1 knickpoints cluster equally well in $\chi$ and $\tau'$, but generation 2 knickpoints but show a large amount of scatter for both metrics (Fig. 8). Similar to $\chi$, knickpoint $\tau'$ shows an inverse correlation with $k_{sn}$ of the next downstream reach (Fig. 8c). This correlation appears non-linear above $k_{sn}$ values of ~ 200 $m^{0.9}$ where there is little difference in $\tau'$.  This suggests that the knickpoints propagate very

slowly when rivers steepen beyond ~ 200 $m^{0.9}$ and their celerity normalized to upstream drainage area increases more rapidly below this steepness threshold. One might be tempted to assign the inverse relationship and spread in $\tau'$ versus $k_{sn}$ data to local variations in rock type, as weaker rocks will have a faster response time (larger $\tau'$ values) and higher steepness (provided other factors, such as uplift rate and climate, are constant) relative to stronger rocks. However, when rock type is included on the $\tau'$-$k_{sn}$ plot, no clustering of $\tau'$-$k_{sn}$ is noted for a particular rock type (Fig. 8d). While somewhat

confounding, this relationship supports our earlier assertion that rock type plays an important, but secondary role in controlling fluvial processes in south-central Crete.

Beyond local variations in rock type we further explore possible explanations for the unpredictability of knickpoint travel distances in south-central Crete per figure 8. The goal of this discussion is to attempt to reconcile our observations in the context of our current understanding of river incision theory and landscape evolution. In the following sections we

discuss the potential for changes in the mechanics of river incision for steep channel segments, changes in river network topology (drainage area exchange and river capture), variable origin for generation-2 knickpoints, break down of assumed drainage area discharge scaling due to karst hydrology found through Mediterranean regions, and the role of chemical weathering in river incision processes. Importantly, these processes might also help explain why the empirically derived value of $n$ is less than 1 in south-central Crete.





### 5.5 Potential factors complicating river profile analysis in south-central Crete

#### 5.5.1 Over-steepened river channels

River channel segments across the study area that reach or exceed $k_{sn}$ values of ~ 300 m$^{0.9}$ often contain vertical waterfalls (Fig. 9). These river sections are only associated with the lowest footwall reaches and are noted in basins 3, 6, 7, 8 and 9 in the Asterousia and basins 13, 16, 17 and 18 in the Dikti. All of the waterfalls occur in Mesozoic carbonate bedrock that is massive or has bedding that is gently dipping upstream. One possible explanation for why these steep river channel segments appear to incise less efficiently than lower gradient channels (e.g. $n < 1$) is that the mechanics of river incision change when vertical waterfalls form (Lamb et al., 2007; DiBiase et al., 2015). Previous research has shown that waterfall formation may either enhance or inhibit river incision rates and would result in an apparent $n > 1$ or $n < 1$, respectively. We suggest that in south-central Crete, carbonate rocks promote the formation of waterfalls at rapid uplift rates. When waterfalls form, the efficiency of river incision is reduced and knickpoints migration rate slows. Continued rock uplift and associated base level fall over-steepens river segments. This process can help explain why generation-2 knickpoints are not observed in basins 6 – 9; river channel segments have been over-steepened such that a change in rock uplift rate is insufficient to generate a new, lower knickpoint. If correct, this interpretation means that quantitative information on the timing of uplift events and uplift rates from river profiles can be obscured or erased from the river network as a result of changing incision processes due to waterfall formation (DiBiase et al., 2015).

#### 5.5.2 Drainage area exchange, river capture, and variable knickpoint origins

To assess the role of drainage area exchange on the fluvial geomorphology of south-central Crete we map the transformed distance variable $\chi$ (Fig. 10). Inspection of $\chi$-maps suggests that there is abundant evidence for drainage basin instability in south-central Crete (Fig. 10a, b). Contrasts in $\chi$ are noted along local divides in both the Asterousia and Dikti ranges (Fig. 10a, b). In the Asterousia the largest $\chi$ contrast is noted in the west where it appears that basins 1 and 3 are gaining drainage at the expense of basin 2 (Fig. 10a). Also noteworthy is basin 7, which is apparently expanding by consuming its neighbor's drainage area (Fig. 10a).

In the Dikti there are strong $\chi$ contrasts along local divides in many of the basins (Fig. 10b). Basins 14 and 15 have very high $\chi$ at their channel heads relative to their neighbors, suggesting they are losing drainage area (Fig. 10b). This interpretation is supported by the elongate and narrow shape of basins 14 and 15 that are widest in the middle and narrow upstream (Fig. 10b). Basin 13 is interpreted as expanding in size at the expense of neighboring basins. This basin has also captured a formally internally-drained basin and some beheaded channels now surround its local water divide (Fig. 10c). When the $\chi$-plots for basin 13 and 14 are compared, it is noted that the elevations for the trunk channel in basin 13 are always higher than elevations in basin 14 for a given $\chi$ distance (Fig. 10d). This elevation shift in a $\chi$-plot is a hallmark of drainage area exchange, where the river elevation for the basin gaining drainage area (the aggressor) shifts to lower $\chi$ for a



given elevation relative to the basin losing drainage area (the victim) (Fig. 10d; Willett et al., 2014). Basin 19 is interpreted as expanding (Fig. 10b). This basin occupies an ancient submarine landslide scar and what remains of the deposit (Alves and Lourenço, 2010). The morphologic legacy of this submarine landslide has not completely been erased from the landscape and favors the expansion of basin 19 because its relative base level is lower than its neighbors at a given distance from the

river mouth. Numerous other local divides appear to be moving and their relative directions are shown on figure 11.

Divide motion, drainage area exchange and river capture can wreak havoc on river profile analysis, destroying the linkages between empirically derived parameters and river incision theory as cast in Eq. (4) and Eq. (5). Although generally implicit, one of the fundamental assumptions in river profile analysis is that the drainage network has remained static trough time. If this assumption is not met, empirical measurements of $k_s$ or $k_{sn}$ cannot be directly linked to the quasi-physical

description of river incision expressed by the stream power incision model. Knickpoint migration speeds will vary in ways not predicted by the knickpoint celerity model, which also requires that drainage area has remained the same during the time when migrating knickpoints are present in a given catchment. Rivers that grain drainage area will appear to steepen as they shift to lower $\chi$ and knickpoint migration rates will increase relative to initial conditions. For basins that loose drainage area, both $k_s$ and $k_{sn}$ will decline and knickpoint migration rates will slow with respect to the initial state of the basin.

When a river capture occurs, a mobile knickpoint will form that is not directly related to changes in the rate of tectonic uplift (e.g. Yanites et al., 2013). While we think that large scale drainage captures are rare in south-central Crete, it is possible that basin 13 represents an exception. The lower generation-1 knickpoint may be related to capture of a formally internally drained basin (Fig. 10c). This potential capture could have been facilitated by an increase in uplift rate due to linkage of the South-Central Crete fault. In such a case the differences in channel steepness above and below the knickpoint

at least partially record the history of relative base level fall rates, but quantitative information on uplift rates have been lost due to changes in drainage area through time. We suggest that much of the noise observed in the generation-2 knickpoint data is due to drainage area loss or gain across the study area. We also acknowledge the possibility that that some of the knickpoints might be non-tectonic in origin (e.g. river captures). Nonetheless, the consistent patterns in knickpoint elevation and $k_{sn}$ for the different river reaches along the coastline leads us to conclude that the majority of the knickpoints in this

study are the result of temporal changes in the rate of rock uplift and that the changes in channel steepness mostly record the *relative* changes in rock uplift rate through time.

**5.5.3 Other possible mechanisms for odd river profile behavior**

Based on observations and analysis of the river systems in south-central Crete the formation of vertical waterfalls and drainage area exchange play an important role in forcing rivers to deviate from expectations of the stream power incision

model. However, there are several other possible processes that could play a role in anomalous behavior of river systems in regions geologically and climatically similar to Crete. While we do not have the data to test these hypotheses, for the sake of completeness we include a brief discussion of each below.



### 5.5.3.1 Karst Hydrology

Carbonates or other highly-fractured and transmissive rocks units subject to the formation of karst have the potential for rapid surface water infiltration that can result in a breakdown of the typically observed power-law relationship between contributing drainage area and water discharge (O'Driscoll and DeWalle, 2006; Allan and Castillo, 2007). The loss of surface water to groundwater in the headwaters of a catchment can result in a reduction in both discharge and river incision rate at higher elevations. At lower elevations where groundwater is delivered back into the surface river network, discharge will be elevated and incision rates higher, relative to areas upstream (O'Driscoll and DeWalle, 2006). The development of efficient groundwater systems can also result in enhanced physical and chemical erosion where ground water springs discharge into the surface water fluvial system (e.g. Dunne, 1990). Groundwater sapping acts to undercut steep river sections resulting in oversteepened river reaches. Groundwater sapping erosion is observed in locations like the Grand Canyon, USA and the Big Island of Hawaii and results in over steepened river reaches (Laity and Malin, 1985; Kochel and Piper, 1986).

Because carbonate is so prevalent beneath Crete specifically, and the circum-Mediterranean region generally, and there is ample evidence of karstic depressions throughout the island (e.g. Rackham and Moody, 1996), we suggest that groundwater hydrology likely plays an important role in the development of river networks here. If groundwater sapping is an important process in the carbonates, assumptions about the scaling between drainage area and discharge made in derivation of the stream power incision model are violated. Groundwater sapping may contribute to the oversteepened reaches and the formation of waterfall in the carbonate units in south-central Crete. Anecdotal evidence exists to support this hypothesis as springs are often observed at the base of waterfalls throughout the study area. Springs are delineated by the presence of vegetation at the base of the vertical waterfall in Figure 9b, for example.

### 5.5.3.2 Sediment flux and grain size distributions

Numerous studies emphasize the important role that sediment plays in river incision processes (Sklar and Dietrich, 2004; Finnegan et al., 2007; Whittaker et al., 2007; Cook et al., 2013; Attal et al., 2015; Shobe et al., 2016). Theoretical models developed to describe this process typically only describe the flux of sediment moving through the system, ignoring the potentially important effects of grain size on incision process (Tucker and Whipple, 2002; Tucker, 2004; Gasparini et al., 2006). Some recent studies have tried to better understand the role of sediment grain size distributions on incision processes (Shobe et al., 2016; Brocard et al., 2016). What has been shown in these latter studies is that sediment grain size distributions affect the relative efficiency of incision along a river profile. Brocard et al. (2016) demonstrated that highly weathered uplands above knickpoints only contributed sand and fine grain sizes to the river networks along the southern flank of the Luquillo Mountains, Puerto Rico. Based on beryllium-10 derived erosion rates and knickpoint celerity modeling, Brocard and colleagues show that knickpoint retreat rates were slower than expected for general incision schemes. It was argued that the small grain sizes derived from highly weathered uplands were ineffective at eroding the lip of knickpoints, slowing retreat rates relative to incision models that suggest knickpoint retreat rates should scale with drainage area.





Some of the areas above generation-2 knickpoints in south-central Crete erode into areas that are not highly weathered, but are underlain by fine-grained marine or terrestrial sediments that are not mapped at the scale of the geologic maps used in this study. These fine-grained sediments are most prevalent in the headwaters of basins 6 – 9 (Fig. 5). Following the rational of Brocard et al., (2016), we suggest that this fine-grained sediment may decouple knickpoint retreat velocity from a simple power-law scaling relationship for some of the steams in south-central Crete. If correct, this mechanism can help explain some of the scatter observed in the generation-2 knickpoint data, as well as the over-steepening of channels below these knickpoints.

### 5.5.3.3 Chemical weathering and river incision

Chemical weathering of river channels has been argued to influence incision processes (Hancock et al., 2011; Murphy et al., 2016). Most studies that investigate the role of chemical weathering on incision processes have looked at its effects on cross-sectional channel width (Hancock et al., 2011). However, it is also possible that chemical weathering could be important in influencing the relative efficiency of erosion along a river channel (Murphy et al., 2016). A parcel of rock exhuming trough a more slowly eroding river reach will interact with the hydrosphere, atmosphere and biosphere for a longer period of time relative to rock exhuming through a more rapidly eroding river segment. These interactions should act to weaken rock through weathering, such that the bedrock in most slowly eroding river reaches will be more easily eroded than bedrock in rapidly eroding channel segments. If it is assumed that the erodibility is uniform along the river profile when it is in fact inversely related to incision rate, the effect will be an apparent slope exponent, $n$, that is less than one. While we currently have no evidence to support this hypothesis, we also do not have any data to discount it. Nonetheless, it remains another possible complicating factor that can affect river incision processes and obscure signals of interest (tectonics and climate) in river profile analysis.

### 6 Study Implications

The implications from this research are important for studies that use river profile geometries to infer tectonic signals. One promising aspect of this study is that we show that the relative spatial and temporal patterns of uplift can be recorded in river profiles consistent with numerous other studies ((Snyder et al., 2000; Kirby and Whipple, 2001, 2012; Wobus et al., 2006; Whittaker et al., 2008; Boulton and Whittaker, 2009; Pritchard et al., 2009; Whittaker and Boulton, 2012; Perron and Royden, 2013; Royden and Perron, 2013; Goren et al., 2014; Whittaker and Walker, 2015). However, other processes, such as waterfall formation, drainage area exchange, karst hydrology and ground water sapping, the role of sediment in river incision, and chemical weathering, may obscure quantitative predictions about time-scales of landscape adjustment and uplift rate history from the modern topography (Prichard et al., 2009; Roberts et al., 2013; Goren et al., 2014). This study emphasizes the fact that measurements made from river profile geometries are completely empirical. In other words, any incision model calibrated using river profiles represent *apparent* values rather than actual stream incision parameters. A good example of this comes from our empirical calibration to derive the slope exponent in this study. While it





is possible that river incision processes are best described by an $n < 1$, it is equally likely that this is solely an artifact of assumptions made in our analysis, namely that the incision process along a river reach can be approximated by constancy in drainage area, local channel slope, and $K$. This study represents a cautionary tale for river profile analysis, emphasizing the care that must be taken when using fluvial topography to infer tectonic signals, particularly in a quantitative sense. Our

results show that rivers in south-central Crete do not respond to tectonic forcing in a manner consistent with the stream power incision model, particularly with respect to river response times as noted by the failure of the knickpoint celerity model. It is important to consider the assumptions embedded in river profile analysis before interpreting results from river profile analysis, including but not limited to power-law scaling between discharge and drainage area, power-law scaling between channel width and discharge, and stasis in drainage area through time. Similarly, it is important to be cognizant that

the quantitative interpretation of river profile geometries used to infer tectonic signals relies on the ability of the stream power incision model to describe the incision process of the channel(s) of interest. If incision processes deviate from the stream power incision model, accurate interpretation of results may be obfuscated because the response time and behavior of the river is challenging to constrain.

## 7 Conclusions

The spatial and temporal changes in the normalized channel steepness index ($k_{sn}$) are consistent with the growth and linkage of two large normal fault systems, the Ptolmey and South-Central Crete faults (Fig. 5c-e, Fig. 6). Further, it is illustrated that river profile segments interpreted to be adjusted to the modern uplift rate, as deduced from studies of marine terraces and are positively correlated to rock uplift rate (Fig. 7). This analysis confirms the findings of numerous other studies that show that river profiles will adjust their gradient to keep pace with the rate of rock uplift (Snyder et al., 2000;

Kirby and Whipple, 2001). While, these findings are promising, we also show that when we empirically calibrate the steam power incision model for south-central Crete the slope exponent, $n$, is less than one. This finding is at odds with most other studies of river incision that typically show $n \geq 1$ (Ouimet et al., 2009; DiBiase et al., 2010; Kirby and Whipple, 2001, 2012; Harel et al., 2016), and suggests, rather counterintuitively, that river incision becomes less efficient when river channels steepen. Knickpoints separating river segments of different steepness in this study area do not migrate in a fashion consistent

with knickpoint retreat rates based on the stream power incision model and a fixed drainage network size, suggesting that other processes are affecting knickpoint migration rates. Evidence exists for the association of oversteepened river reaches and the formation of vertical waterfalls (Fig. 9) and for the exchange of drainage area between adjunct basins and the capture of formally internally drainage basins (Fig. 10). Both of these processes can help explain the development of oversteepened river reaches and the breakdown of scaling between knickpoint migration distance and modern upstream contributing

drainage area. Other processes that may contribute to the anomalous behavior of rivers in south-central Crete include karst hydrology and groundwater sapping, the unaccounted role of sediment flux and grain sizes on river incision, and linkages between chemical weathering and the efficiency of river incision. Our research represents a cautionary tale for river profile



analysis, particularly when river profile geometries are used to quantitatively invert for rock uplift histories, as the numerous assumptions made to produce data from river profiles may not hold in particular settings. As such, parameters derived from river profiles that are used to infer stream incision histories should be considered apparent, rather than actual.

## 8 Code availability

5   All of the river profile analysis codes are available upon request from the corresponding author.

## 9 Author contributions

S.F.G. and K.W.W. conceived and designed the study. S.F.G. wrote the data analysis codes and conducted all analyses. S.F.G. and K.W.W. contributed equally to interpretation of results and writing the manuscript.

## 10 Competing interests

10   The authors declare that they have no conflict of interest.

## 11 Acknowledgements

Acknowledgment is made to the Donors of the American Chemical Society Petroleum Research Fund for primary support of this research under grant #50792-DNR8. Additional support was provided by a 2010 Geological Society of America Graduate Student Research Grant (number 9596-11) and a 2012 Sigma Xi Grant-in-Aid of Research to Gallen. We would also like to thank the Institute of Geology and Mineral Exploration in Greece for granting us permission to carry out field studies on Crete. Nicholas Panzera is thanked for digitizing the geologic map of south-central Crete. We are indebted to Charalampos Fassoulas and Nikos Somarakis for logistical support while in the field.

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



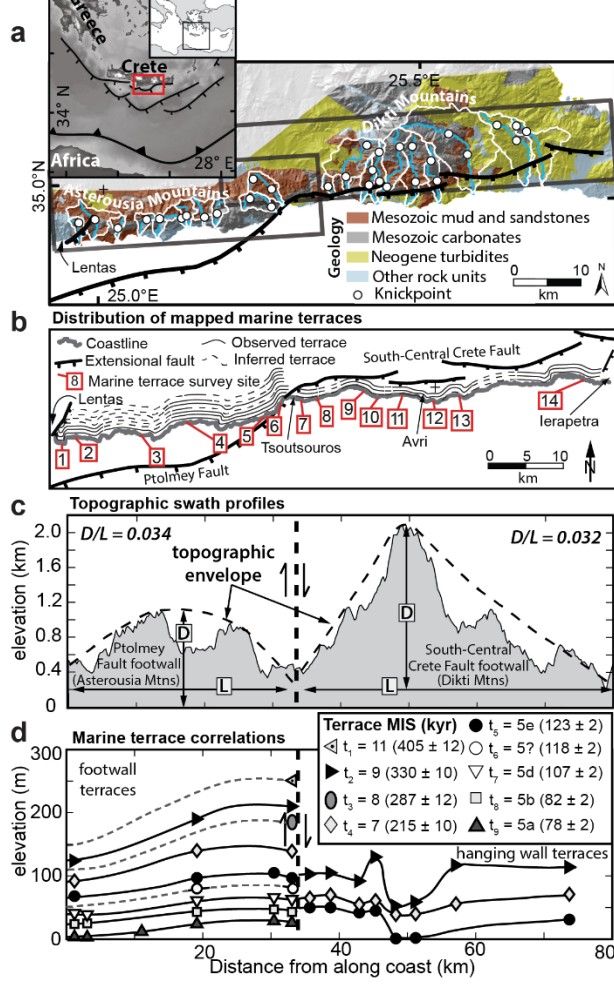

**Figure 1:** Inset maps show the location of the study area in south-central Crete and generalized tectonic setting. **a.** Digitized 1:50,000-scale geologic maps (Greek Institute of Geology and Mineral Exploration, 1972; 1974; 1977; 1987) with the outlines of the 21 drainage basins in thin white lines and trunk river channels in light blue. The dark gray rectangles show the locations of the elevation profiles shown in panel **c**. Active normal faults are shown as black lines with the hatch marks on the hanging wall after Gallen et al. (2014). **b.** Map of the south-central coastline of Crete with the geographic distribution of Pleistocene marine terraces. The figures are not drafted to scale, such that the vertical separation between terraces is relative. Sites 1 – 14 are the marine terrace survey sites from Gallen et al. (2014). The locations of some of the major towns and faults are shown. **c.** Maximum topography of south-central Crete as determined by swath profiles taken along the Asterousia and Dikti Mountains (see figure 1a for location). Dashed lines indicate the profile of maximum displacement as determined by correlation of the highest peaks in the respective fault blocks. The thick vertical dashed line indicates the location and sense of motion for an active extensional fault, the South-Central Crete fault. **d.** Terrace correlations for the south–central coastline of Crete (Gallen et al., 2014).





Earth **Surface**
Dynamics
Discussions

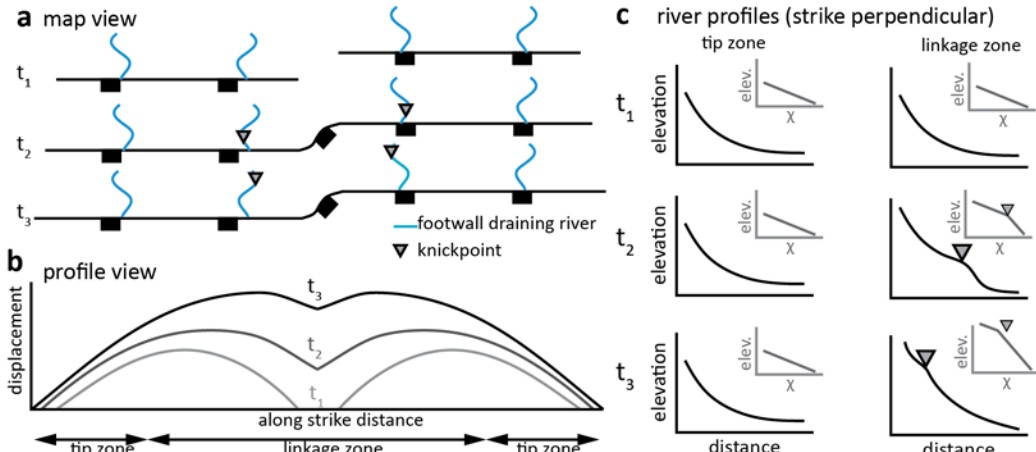

**Figure 2:** Conceptual model for river profile response to growth and linkage of adjacent normal faults. **a.** map view of linkage history. Knickpoints form near the linkage zone due to the rapid increase in uplift rate fallowing linkage of two previously independent faults. **b.** along strike profiles showing the cumulative displacement history resulting from linkage of the normal faults. **c.** the predicted response of footwall river profiles in the tip and linkage zones. Rivers in the tip zone will experience little change in relative uplift rate; whereas those in the linkage zone will experience a rapid increase in the rate of uplift. The increase in uplift rate will result in the formation of a knickpoint that migrates upstream over time. The knickpoint is identified as a break in slope in a plot of the transformed distance variable, $\chi$, versus elevation.



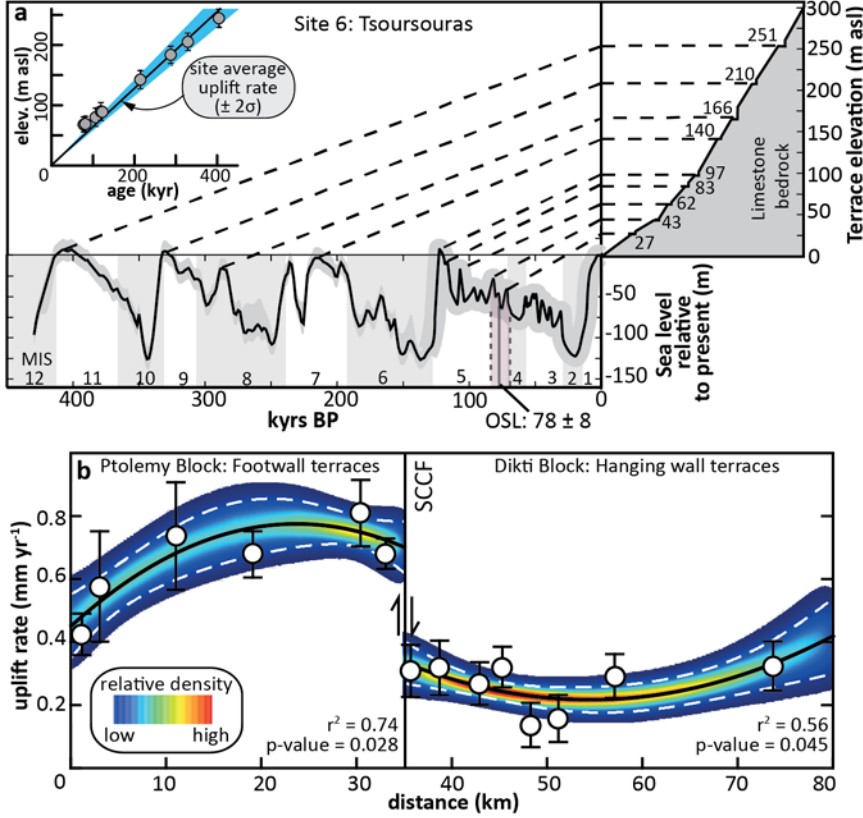

**Figure 3: a.** Correlation of the Tsoutsouros (**a** – site 6; figure 1b) terrace sequences to the Late Quaternary global sea level curve (after Gallen et al., 2014). The 0 to 125 kyr sea level curve was compiled from Lambeck and Chappell (2001), while the 125 to 450 kyr segment is from Waelbroeck et al. (2002). Marine isotope stage (MIS) boundaries are from Lisiecki and Raymo (2005). Each terrace sequence is anchored to the sea level curve with an optically simulated luminescence (OSL) burial age of fine sand collected from marine terrace deposits. The Tsoutsouros sequence shows the OSL age of the lowest terrace in the sequence from Gallen et al. (2014). Inset diagrams show the total uplift versus estimated terrace age ($\pm 2\sigma$) and site average uplift rate from the Monte-Carlo regression analysis (dashed back line) with $2\sigma$ uncertainties (blue polygon). **b**. Site average uplift rates ($\pm 2\sigma$) are displayed as white circles. The coastal uplift model is shown as the black line and the white dashed lines depict the $\pm 2\sigma$ uncertainties. The colored background shows the relative density of regression paths from the Monte-Carlo analysis. The location and sense of motion of the South-Central Crete fault (SCCF) is shown.





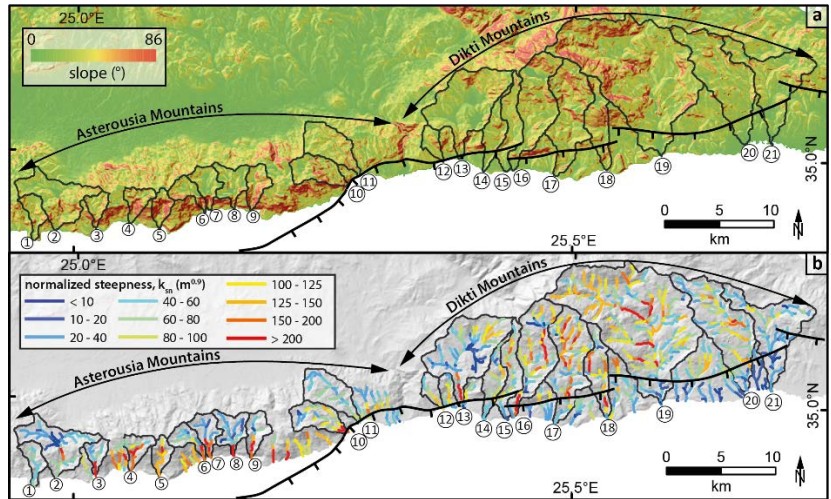

**Figure 4: a.** map of the slope distribution overlain with the location of active faults and the 21 drainage basins analyzed in this study (black polygons). **b.** normalized steepness index ($k_{sn}$) map of the channels that drain $> 0.1$ km$^2$ area for the 21 basins investigated in this study.



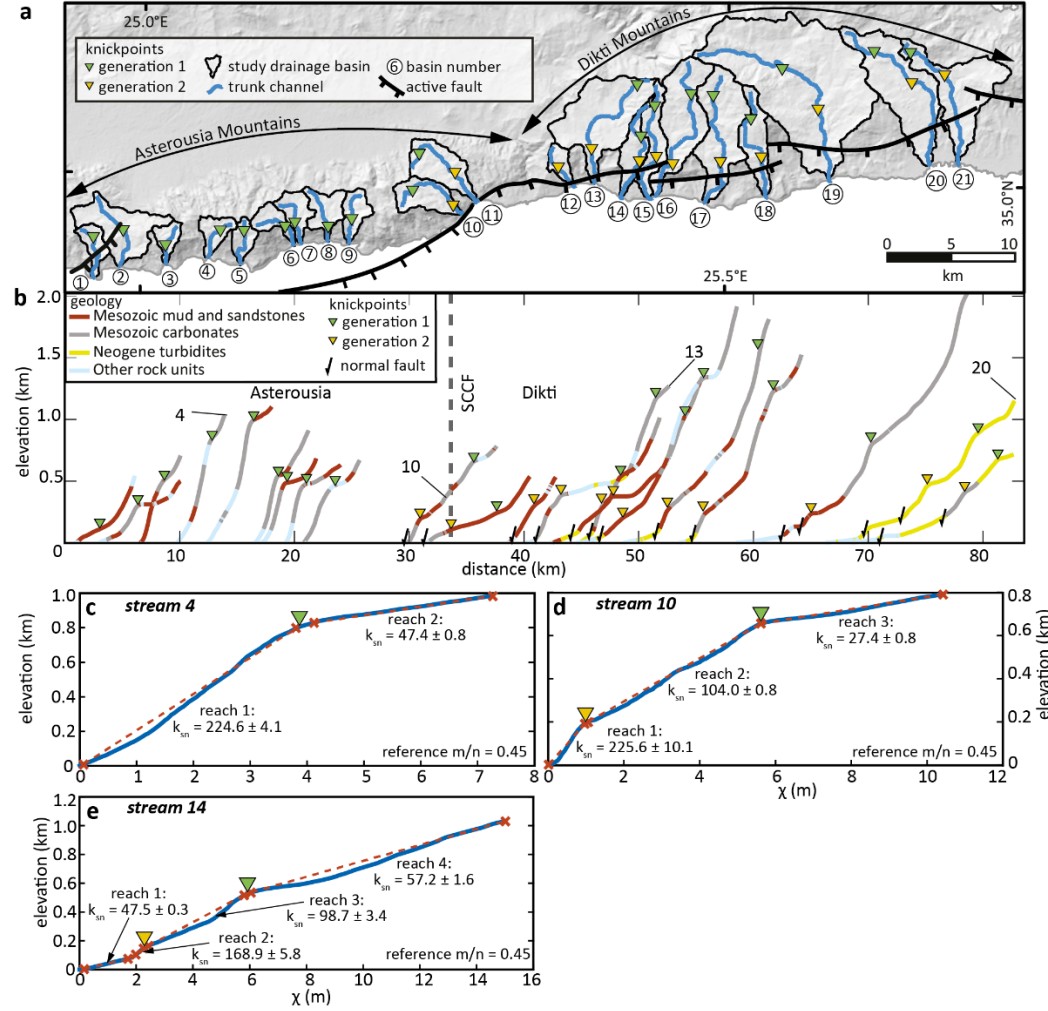

**Figure 5: a.** shaded relief map showing the location of the 21 drainage basins, trunk river channels and fluvial knickpoints identified in this study. **b.** trunk channel river profiles for the 21 basins of this study plotted versus outlet distance from the town of Lentas. The rock-type over which each river flows is shown as well as the major fluvial knickpoints identified during this study. **c-e.** Typical examples of $\chi$ versus elevation plots for rivers in south-central Crete (thick blue lines). Knickpoints are shown as triangles and the dashed red lines show the linear regressions used to determine reach specific normalized channel steepness index ($k_{sn}$).





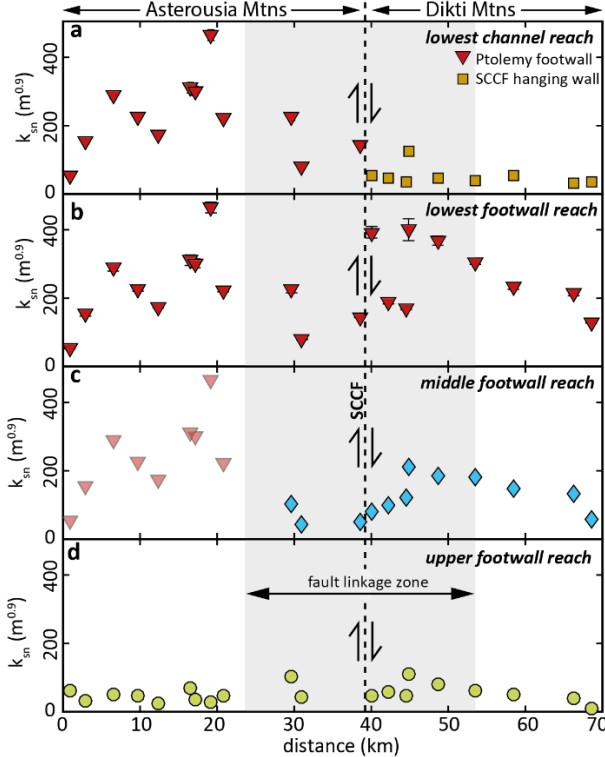

**Figure 6:** Normalized steepness index ($k_{sn}$) for channel segments bound by knickpoints. The location where the South-Central Crete fault (SCCF) separates river channels in the footwall and hanging wall is shown as the vertical dashed line. **a.** data for the lowest river reaches. Note that for basins 1 – 12 the lowest reach is in the fault footwall, whereas for basins 13 – 21 the lowest reach is in the fault hanging wall. **b.** the normalized steepness index for the lowest river reaches in the both footwalls. **c.** data for the middle reach for rivers that contain two knickpoints (basins 10 – 12) and lowest reach for rivers with only 1 knickpoint (basins 1 – 9). **d.** the normalized steepness index for the river reaches above the highest knickpoints.



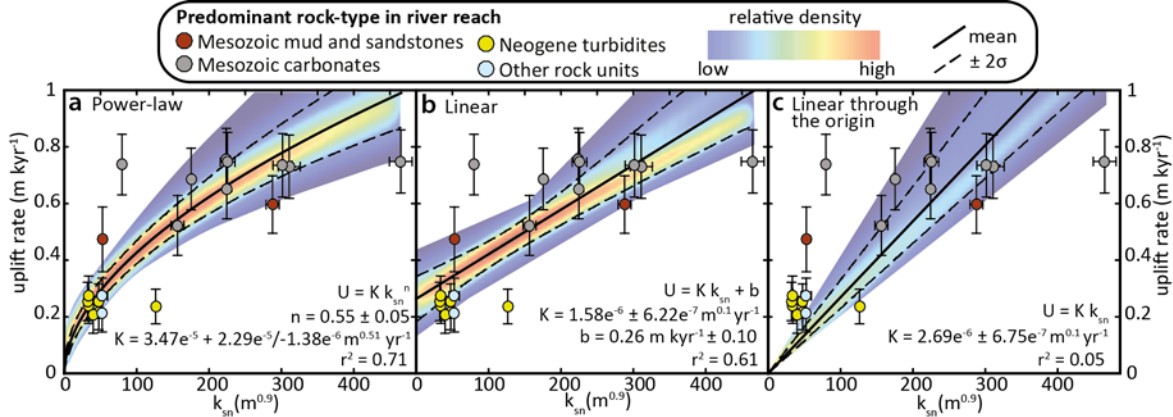

**Figure 7:** Plots of the normalized steepness index ($k_{sn}$) for the lowest river reaches (fig. 6a) versus modeled uplift rate from marine terrace (fig. 3b). The predominant rock-type underlying each river reach is noted. Each panel shows the mean (solid black line), $\pm 2\sigma$ uncertainties (dashed black lines), and relative path density (colored background) from a bootstrap regression analysis using 1000 trials. Regressions are used to calibrate parameters in the stream power incision model under different assumptions and are only valid based on the condition that each river reach is eroding at the same rate as local rock uplift. **a.** A power-law regression, **b.** unconstrained linear regression, and **c.** linear regression forced through the origin. All parameter uncertainties are presented as $\pm 2\sigma$.





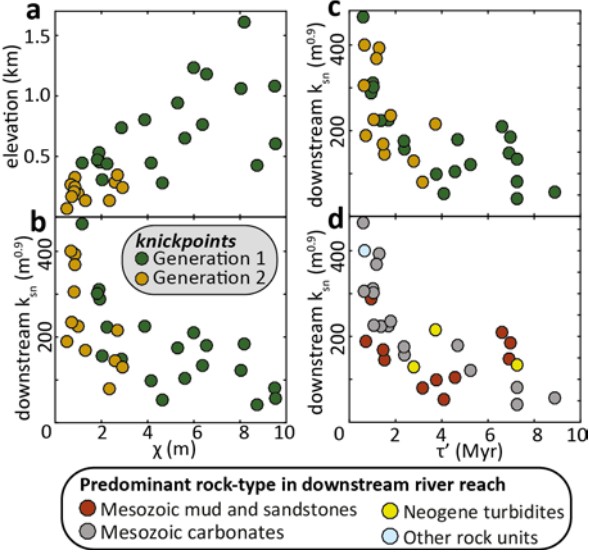

**Figure 8:** Plots of major knickpoint metrics from trunk channels in south-central Crete. **a.** $\chi$ versus elevation for generation-1 and 2 knickpoints. The stream power incision model predicts knickpoints from a common origin will remain at the same $\chi$ as they travel upstream provided that n = 1 and erodibility and drainage area are constant in space and time. **b.** $\chi$ versus $k_{sn}$ for generation-1 and 2 knickpoints. Note that for each generation there is an apparent weak inverse correlation between $\chi$ and $k_{sn}$. **c.** $\tau'$ versus $k_{sn}$ for generation-1 and 2 knickpoints. **d.** same data as c, but each point is colored according to the predominant rock type on the next downstream reach.





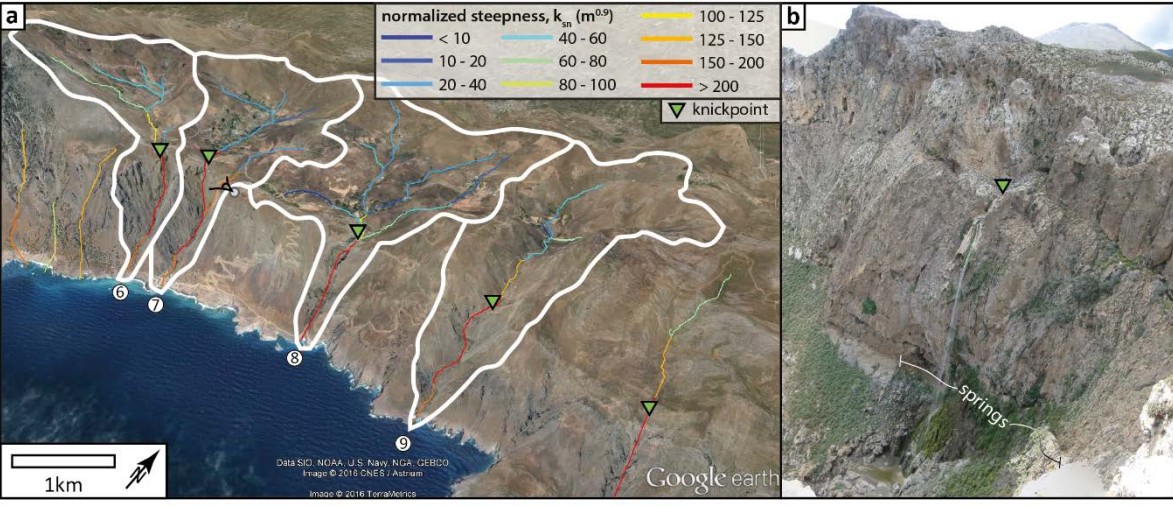

**Figure 9: a.** Perspective Google earth image of basins 6 – 9 (white outlined polygons) with normalized steepness index ($k_{sn}$) of channels draining areas > 0.1 km$^2$. Channels with $k_{sn}$ > 200 m$^{0.9}$ are commonly associated with vertical waterfall reaches. **b.** photograph of vertical waterfall from basin 7. The location of the photo is shown in a. Location of springs is noted.





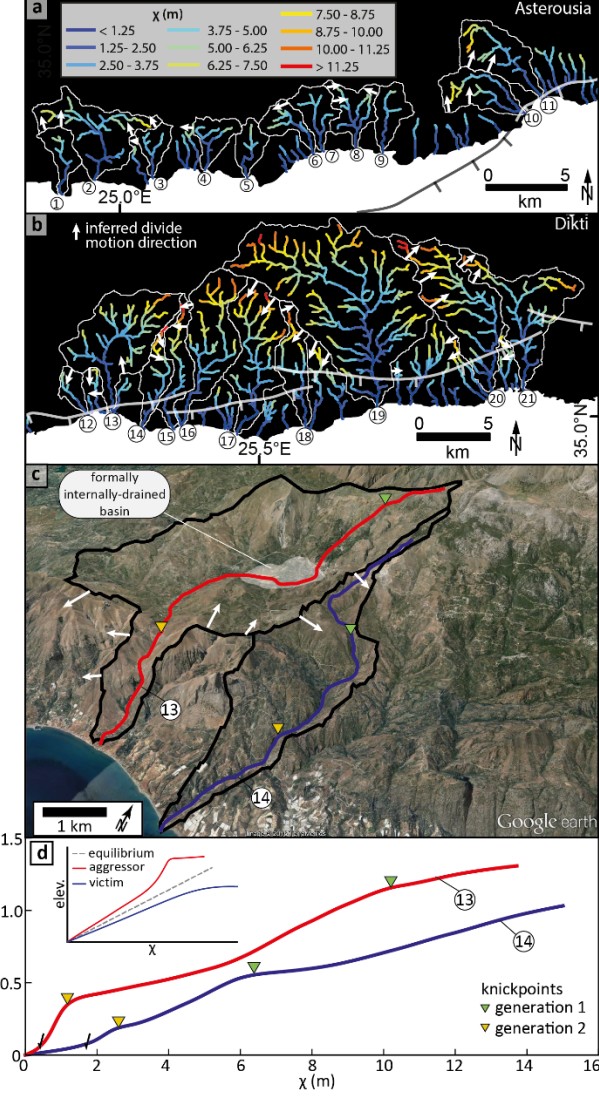

**Figure 10: a, b.** $\chi$-maps of the Asterousia (a) and Dikti (b) mountains for rivers draining areas >0.1 km². Drainage basin divides are noted by the white polygons and the interpreted direction of divide motion, based on local contrasts in $\chi$ across drainage divides, is denoted by the white arrows. **c.** Google Earth image of basins 13 and 14 with the trunk river channels shown by the red and blue lines, respectively. The interpreted direction of divide motion is noted by the white arrows. **d.** $\chi$-elevation plots for the trunk channels in basins 13 and 14. Note the low $\chi$ values for a given elevation for basin 13, relative to basin 14. Inset shows the predicted shifts in $\chi$ for a river gaining drainage area (red – aggressor) and a river losing drainage are (blue – victim).