# Peer review of "River profile response to normal fault growth and linkage: An example from the Hellenic forearc of south-central Crete, Greece"

_Earth Surface Dynamics, 2016_

## Referee Comment (RC1) · S. Boulton (Referee) · 21 Dec 2016

This paper investigates river response to uplift along the southern margin of Crete, using well known fluvial geomorphic metrics and previously published data on the rate of uplift and behaviour of recently linked normal faults. The paper draws some interesting conclusions on the role of river capture on river profile evolution and the potential pitfalls of this technique in some areas. The background to river analysis and the geology/geomorphology of the study area are very well presented and described as are the results. The discussion is well structured and outlines the authors arguments well. There are only minor comments that need to be addressed prior to publication.

Page 3, lines 10-15. This paragraph sets up an argument based upon the work of Attal

et al., (2015) and Shobe et al., (2016). Shobe et al., is described but the work of Attal is not elaborated upon and instead Brocard et al., (2016) is introduced. I find that this is rather unsatisfactory as Attal et al.'s contribution is unclear in this review. Page 7, line 6. Missing parenthesis after (Fig. 2. Page 7, Lines 10 and 11. See also Kent et al., (2016). Kent, E., Boulton, S. J., Whittaker, A.C., Stewart, I.S., & Alçiçek, M.C. 2016. River profiles as recorders of fault linkage and slip rate increases in the Gediz (Alaşe-hir) Graben, Turkey. Earth Surface Process and Landforms. Doi: 10.1002/esp.4049 Page 9, line 19. Headward Section 3.1/4.1. These sections incorporate a monte carlo approach to calculate the uplift rate along the coast and at individual sites, a nice idea but the results leave me with a number of questions/comments. An average uplift rate is determined, shown in figure 3a (I think the figure caption should state the uplift rate or it should be shown on the figure). However, it needs to be made clearer that these are post-linkage uplift rates. Also I would have liked to see the authors try to narrow down the timing of linkage, as currently they simply use the previous estimate of < 1 Myrs ago. Also have you any constraints on pre-linkage uplift rates? Although the mouth of the rivers on the Dikti block enter the sea on the hangingwall block, knickpoints in these rivers are still going to be formed by the initiation of faulting or change in footwall uplift rates on the SSCF. How do the rates of hangingwall and footwall uplift compare? Many studies of the hangingwall to footwall motion cite ratios of $\frac{1}{4}$ to 1/3 partitioning. Page 12 section 3.3 I am interested that you have defined knickpoints as a 25% difference between Ksn upstream and downstream, what is your rationale for this number? Is that consistent with where known active faults cross channels? Section 4.3 Although I agree that the two sets of knickpoints represent two phases of development it would be nice if there was some test of this hypothesis. How about presenting distance migrated upstream vs catchment drainage area. Faults of the same generation should exhibit a power law relationship. Section 5.1 lines 18-21. Ah – I think that this information on the timing of fault linkage should be presented earlier. Page 15, Line 31 – . . .in uplift rate determinations. . .? Page 16, line 21 (also page 17, line) What mechanism caused the first increase in uplift rate, if there is no linkage? What evidence is there for previous

**ESurfD**
slow uplift? Why does a shallow river indicate slow uplift? Why could not the upper knickpoint represent the initiation of faulting? Page 18, lines 29-32. I might have misunderstood but this sentence appears to contradict the discussion two pages earlier, as you are now saying the upland areas are 'relict' topography from prior to fault initiation. Page 19, lines 23. I know that assumptions need to be made, but having been to gorges in southern Crete, channel narrowing seems to be important and should not be discounted so easily. Perhaps saying that this variable is beyond the scope of the paper would be better than saying it is not important. This change might also explain some of the variability you observe in your data. Page 20, line 24. Whittaker and Boulton also (2011) demonstrated that knickpoint migration is a function of uplift rate, with higher uplift rates resulting in more rapid migration of knickpoints through the landscape.

---

## Author Comment (AC1) · 31 Dec 2016

**This paper investigates river response to uplift along the southern margin of Crete, using well known fluvial geomorphic metrics and previously published data on the rate of uplift and behaviour of recently linked normal faults. The paper draws some interesting conclusions on the role of river capture on river profile evolution and the potential pitfalls of this technique in some areas. The background to river analysis and the geology/geomorphology of the study area are very well presented and described as are the results. The discussion is well structured and outlines the authors arguments well. There are only minor comments that need to be addressed prior to publication.**

*We thank Dr. Boulton for her thorough review and below we detail how we have addressed each of her specific comments in the revised version of our manuscript. Dr. Boulton's comments are in **bold text** and our responses are in* blue italics.

**Page 3, lines 10-15. This paragraph sets up an argument based upon the work of Attal et al., (2015) and Shobe et al., (2016). Shobe et al., is described but the work of Attal is not elaborated upon and instead Brocard et al., (2016) is introduced. I find that this is rather unsatisfactory as Attal et al.'s contribution is unclear in this review.**

*We agree with the reviewer and have elaborated on Attal et al.'s contribution to our understanding of this topic. We have also revised the text so that the Brocard study is better integrated into the paragraph by removing the initial sentence that specifically mentions Attal et al., (2015) and Shobe et al., (2016) because it was unnecessary. The new text is as follows:*

*"Several recent studies highlight observations of spatially heterogeneous changes in the flux and size of sediment delivered to the channel in transient landscapes that might be missed by simple expressions of sediment flux that are only scaled to local channel slope and drainage area, a proxy for discharge. Attal et al. (2015) document that hillslope steepness and erosion rates are positively correlated to changes in sediment grain size on hillslopes and along the adjoining river channel bed in catchments undergoing transient adjustment."*

**Page 7, line 6. Missing parenthesis after (Fig. 2.**

*We have added the closing parenthesis.*

**Page 7, Lines 10 and 11. See also Kent et al., (2016). Kent, E., Boulton, S. J., Whittaker, A.C., Stewart, I.S., & Alçiçek, M.C. 2016. River profiles as recorders of fault linkage and slip rate increases in the Gediz (Ala¸sehir) Graben, Turkey. Earth Surface Process and Landforms. Doi: 10.1002/esp.4049**

*Thank you for pointing us toward this relevant study. We have read this new research and have included the reference into our manuscript.*

**Page 9, line 19. Headward**

*We have corrected this spelling error.*

**Section 3.1/4.1. These sections incorporate a monte carlo approach to calculate the uplift rate along the coast and at individual sites, a nice idea but the results leave me with a number of questions/comments. An average uplift rate is determined, shown in figure 3a (I think the figure caption should state the uplift rate or it should be shown on the figure). However, it needs to be made clearer that these are post-linkage uplift rates.**

*We have added a sentence to the figure caption indicating that the site average uplift rate is for site 6. To better clarify that figure 3b shows post-linkage uplift rates we have changed the caption to state "b. Post-fault linkage site average uplift rates…".*

**Also I would have liked to see the authors try to narrow down the timing of linkage, as currently they simply use the previous estimate of < 1 Myrs ago.**

*In the first section of the discussion (section 5.1 Long-term versus intermediate vertical displacement patterns) we discuss the rational that we use to approximate the timing of linkage. Our rational for placing this calculation so late in the text is because it is an interpretation and would be difficult to explain elsewhere in the text.*

**Also have you any constraints on pre-linkage uplift rates?**

*We do not have any independent constraints on the pre-linkage uplift rates. We had initially hoped that we would be able to: (1) use the post-linkage uplift rates and lowest river profile segments that cross the normal fault footwalls in order to empirically derive variables in the stream power incision model. (2) With a calibrated incision model, we would then be able to infer pre-linkage uplift rates from the middle river profile segment (the segment between the lower and higher knickpoints) from each footwall. However, for reasons that become clear in the manuscript, we lack confidence in our empirical calibration of the stream power model for south-central Crete, and thus decided not to infer uplift rates from the river profiles.*

**Although the mouth of the rivers on the Dikti block enter the sea on the hangingwall block, knickpoints in these rivers are still going to be formed by the initiation of faulting or change in footwall uplift rates on the SSCF. How do the rates of hangingwall and footwall uplift compare? Many studies of the hangingwall to footwall motion cite ratios of 1/4 to 1/3 partitioning.**

*We do not have independent constraints on the rates of uplift on the Dikti footwall; however, we can compare the post-linkage uplift rates from the hanging wall and footwall of the composite fault system near the site of linkage (see figure 3b). This analysis was initially reported in Gallen et al. (2014), in which we published our marine terrace chronostratigraphy and rock uplift results that we utilize as a starting point for this manuscript. In Gallen et al. (2014), we found that the hanging wall-to-footwall uplift rate ratio is roughly 1/3.*

**Page 12 section 3.3 I am interested that you have defined knickpoints as a 25% difference between Ksn upstream and downstream, what is your rationale for this number?**

*This number is admittedly arbitrary, but was sufficiently large to avoid inclusion of small jumps in river channel steepness that are more likely to represent noise or artifacts in the DEM, while also being small enough that we could identify inflections in all river profiles analyzed in the study. To clarify this point we have added the following text:*

*"While this change in steepness is admittedly an arbitrary cut-off to identify knickpoints, it is sufficiently large to avoid including unwanted noise or DEM artifacts in our analysis and sufficiently small so that we could confidently identify inflection points in all river profiles analyzed in this study."*

**Is that consistent with where known active faults cross channels?**

*The jumps in normalized channel steepness ($K_{sn}$) where the channel crosses over active faults are generally higher than 25%. The mapping of active faults in the study area is detailed enough that we are confident that we can identify knickpoints related to spatial changes in rock uplift rate across mapped faults from transient knickpoints resulting from temporal changes in slip rate/uplift rate across said faults.*

**Section 4.3 Although I agree that the two sets of knickpoints represent two phases of development it would be nice if there was some test of this hypothesis. How about presenting distance migrated upstream vs catchment drainage area. Faults of the same generation should exhibit a power law relationship.**

*The suggested distance migrated vs. drainage area analysis was performed before our initial submission. We decided not to include this analysis because the metric χ can be used in much the same way and thus the extra figure provided little additional insight. Provided that substrate erodibility is roughly homogeneous, knickpoints of a common origin will have the same χ value. To clarify the ability of χ to assist in the determination of whether or not knickpoints are of a common origin we have added the following text to the end of section 2.5, our background on river profile analysis: "Furthermore, if substrate erodibility is homogenous, χ represents a measure of the river response time to a state change in incision (uplift) rate that is manifested in the formation and presence of a knickpoint. Knickpoints from a common origin will travel at the same rate in χ transformed space. In other words, knickpoints found at approximately the same χ distance are related."*

*It is worth noting that much of our discussion focuses on testing if these knickpoints are related. The most compelling evidence is the changes in the relative patterns of river channel steepness that are consistent with the geological evidence of recent fault linkage and the predictions of changes in the spatial and temporal patterns of uplift based on fault mechanics theory. Our discussion largely focuses on an attempt to understand why knickpoint travel distances for streams from the south-central coast of Crete do not conform to predictions of the stream power incision model.*

**Section 5.1 lines 18-21. Ah – I think that this information on the timing of fault linkage should be presented earlier.**

*We understand the desire of the reviewer to present this earlier, but we think that this calculation is better suited for the discussion because it is an interpretation. However, if the reviewer feels strongly that we should move this discussion to earlier in the manuscript we can try to accommodate this suggestion.*

**Page 15, Line 31 – . . .in uplift rate determinations. . .?**

*Thanks for catching this typo. The reviewer is correct and we have fixed the typo.*

**Page 16, line 21 (also page 17, line) What mechanism caused the first increase in uplift rate, if there is no linkage? What evidence is there for previous slow uplift? Why does a shallow river indicate slow uplift? Why could not the upper knickpoint represent the initiation of faulting?**

*This is an excellent question. We discuss this exact question at the end of this section. The reviewer brings up a good point about the possible origins of the signal of uplift recorded by the river reaches above generation 1 knickpoints and we address this below.*

**Page 18, lines 29-32. I might have misunderstood but this sentence appears to contradict the discussion two pages earlier, as you are now saying the upland areas are 'relict' topography from prior to fault initiation.**

*Dr. Boulton brings up a good point and based on the previous comment we have revised the second sentence of the last paragraph in Section 5.3.1 for clarification. The revised sentence is as follows: "Our interpretation is that many of these low-gradient reaches possibly record the background regional rates of rock uplift of the forearc prior to faulting and/or the uplift rates at the initiation of normal faulting along the south-central coast of Crete."*

**Page 19, lines 23. I know that assumptions need to be made, but having been to gorges in southern Crete, channel narrowing seems to be important and should not be discounted so easily. Perhaps saying that this variable is beyond the scope of the paper would be better than saying it is not important. This change might also explain some of the variability you observe in your data.**

*The reviewer is entirely correct and we acknowledge that so easily disregarding the influence of changes in channel width is not permitted. To clarify we have modified the end of the first paragraph in section 5.3.2 as follow:*

*"However, it is beyond the scope of this study to assess the impact of this variable on river profile analysis. Nonetheless, our results demonstrate that rivers in south-central Crete steepen in response to increased uplift rates.  Under the assumption that river channel steepening is the dominant mechanism by which rivers respond to changes in rock uplift rate, regression through $k_{sn}$ – uplift rate data by an orthogonal least-squares method can be used to empirically calibrate parameters in the stream power incision model under different assumptions."*

**Page 20, line 24. Whittaker and Boulton also (2011) demonstrated that knickpoint migration is a function of uplift rate, with higher uplift rates resulting in more rapid migration of knickpoints through the landscape.**

*We are aware of the empirical work of Whittaker and Boulton (2011) on the relationship between knickpoint migration rates, climate and uplift. Here, however, we are simply referring to theoretical considerations and have modified this sentence to make this explicit. The revised sentence is as follows: "It is important to note that Eq. (9) does not incorporate a component of uplift, such that from theory alone, knickpoint celerity is not dependent on uplift rate."*

*We note that while Whittaker and Boulton (2011) document a correlation between rock uplift rate and knickpoint travel distance, the exact factor determining those changes in unknown. For example, unaccounted for changes in channel width, sediment flux, sediment grain size, and or rock fracture may change in concert with rock uplift rate. In such a case, this would result in changes in the slope exponent, n, or erodibility parameter, K, in the stream power incision model. In such a case, changes in n or K would be responsible for driving changes in knickpoint celerity, rather than uplift rate, although these parameters might be dependent on rock uplift rates. However, we think that an extended discussion of the work of Whittaker and Boulton (2011) is beyond the scope of this study as we are merely trying to illustrate the link between river channel response time and the erodibility constant.*

---

## Referee Comment (RC2) · M. Ford (Referee) · 12 Jan 2017

This paper presents a detailed analysis of vertical motions on active faults and of river profile evolution in response to normal fault growth and linkage in southern Crete. The paper presents good data sets, is well written and illustrated and gives a thorough review of literature. My comments focus on the structural framework of the study area and the interpretation of data in terms of fault growth and linkage. On these subjects I have concerns and questions regarding the application of models to the case study, terminology used and the derived interpretations.

The objective of the paper is to use uplift history derived from marine terraces and river profile analysis to study the timing and stages of fault growth and linkage (abstract and page 4 lines 17-20, for example) and to examine the behaviour od rivers in response to fault linkage. As presented in Figure 2, standard mechanical models for normal fault growth and linkage demonstrate uplift of the footwall on the two linking fault segments and subsidence in their hangingwalls. Because of the nature of coastal erosion, the southern Crete study integrates uplift data derived from marine terraces from the footwall of the Ptolemy fault with uplift in the hangingwall of the South Central Crete Fault (SCCF). In this case linking uplift rates and stream behaviour with fault mechanics theory is not straightforward and you need to be very clear and careful in your argumentation. It appears that you are comparing the uplift history of the SCCF hanginwall marine terraces directly to footwall uplift on the Ptolemy fault.

A regional component of uplift of the island of Crete appears to be superimposed on vertical motions generated by fault activity, thus exaggerating footwall uplift on both faults (Fig. 1c). When did this regional uplift occur with respect to the fault history? The observed net uplift in the hangingwall of the SCCF indicates that the regional component is greater than the hangingwall subsidence generated by the SCCF as the authors state in page 5, lines 30-33. So it must be a pretty substantial vertical motion! Can you quantify the regional uplift? This regional component should be removed before calculating D on the faults (see below).

Page 1, Line 16-17. 'Fault mechanics predicts that when adjacent faults link into a single fault the uplift rate in the linkage zone will increase rapidly'. You need to specify that you are referring to footwall uplift. To improve clarity in your analyses of uplift a suggestion is to consistently specify whether you are referring to composite footwall uplift or the composite hangingwall uplift.

The data represented in Figure 1c (maximum footwall topography) is derived from the footwall of both faults and shows the classic bell shaped curves typical of two fault segments. As the profiles represent the footwall of both faults the SCCF would not occur between the two profiles as indicated on the graph. It is notable that the footwall uplift on the SCCF is twice that on the Ptolemy fault, implying that it a larger fault, with

perhaps a longer history? The highest topography is further back from the fault also suggesting a longer history of uplift and erosion? Can you comment on this? Is the difference in fault size and age relevant for your interpretation of the river behaviour?

The data in figures 1d (marine terrace correlations) and 3b (uplift rate from marine terraces) are derived from the footwall of the Ptolemy fault and the hangingwall of the South Central Crete Fault. The history of footwall uplift can therefore only be deciphered for the Ptolemy Fault. It is on this fault that you see the clear acceleration in uplift toward the relay zone. I am intrigued by the abrupt eastward termination of these footwall terraces along a NS line inland from stations 6-7. Can you comment on this? Is there a N-S fault here or is the map incomplete? This could be relevant for your interpretation. The mismatch and incoherence in marine terrace behaviour between the two blocks that you discuss as unusual, is in my opinion, because you are passing eastward across the fault from uplifting footwall to subsiding hangingwall. The decrease in uplift toward the centre of the Dikti block is exactly what you would expect in the subsidence profile of a normal fault hangingwall (Fg. 3b). This also fits perfectly with the fault mechanics model predictions when combined with a regional uplift.

P5, section 2.2. I see no evidence for horst structures on the maps you present, ie. I cannot see the conjugate north dipping faults on the north side of the Asterousia and Dikti mountains that would be necessary for these to be horst structures. Your maps indicate that these mountains represent fault block crests in the footwalls of the two studied faults. This would indeed have to be the case for you to apply the fault growth model presented in Figure 2.

Page 13, 4.1: Section 5.1 Calculation of D/L ratio (figure 1). You use maximum topography in the footwall of the two faults as a proxy for displacement and this to calculate D/L ratios for the faults. However observed footwall relief represents only footwall uplift, here combined with the regional uplift component. The throw or vertical component of fault displacement is the sum of footwall uplift plus hangingwall subsidence, a value that you probably don't have, as is often the case. The ratio of long-term footwall uplift

to hangingwall subsidence (U:S) has been estimated in various rifts (e.g. Basin and Range; Stein et al 1988) and by elastic dislocation modelling of uplifted terraces in rifts such as the Gulf of Corinth (King, 1998; Armijo et al. 1996). These studies propose U:S values ranging from 1:6 to 1:1 (see McNeill and Collier 2004, for example). If no local constraints are available most authors use values for U:S of 1:2 or 1:3 to derive a reasonable estimate of throw. The displacement along the fault is then calculated assuming a reasonable dip for the normal fault. However you have the additional problem of having to remove the regional uplift component before doing this calculation. Why is D for the Ptolmey fault measured with respect to sea level while D for the South Central Crete fault is measured with respect to a higher reference level (200m above sea level?)?

As regards the river profile analysis, it is important to emphasise in your interpretations that the western rivers cut across only the footwall of the Ptolemy fault while the eastern rivers cut across both the footwall and hangingwall of the SCC fault. The regional component of uplift should not be forgotten in the interpretation of these rivers.

I conclude by requesting that the authors clarify the complexity of the structural framework of their study area, the true structural position of data sets, and the terminology used. This will probably have an impact on the presentation of interpretation of tectonic signals in river profiles.

Mary Ford CRPG, Nancy

---

## Author Comment (AC2) · 13 Jan 2017

**This paper presents a detailed analysis of vertical motions on active faults and of river profile evolution in response to normal fault growth and linkage in southern Crete. The paper presents good data sets, is well written and illustrated and gives a thorough review of literature. My comments focus on the structural framework of the study area and the interpretation of data in terms of fault growth and linkage. On these subjects I have concerns and questions regarding the application of models to the case study, terminology used and the derived interpretations.**

*We thank Dr. Ford for taking the time to review our manuscript and provide important and useful critical feedback. Below we detail how we have address Dr. Ford's comments, where we have made changes in our manuscript and explain why we respectfully disagree with a specific comment or criticism. Dr. Ford's comments are in* **bold black text** *and our responses are in blue italic text.*

**The objective of the paper is to use uplift history derived from marine terraces and river profile analysis to study the timing and stages of fault growth and linkage (abstract and page 4 lines 17-20, for example) and to examine the behaviour od rivers in response to fault linkage. As presented in Figure 2, standard mechanical models for normal fault growth and linkage demonstrate uplift of the footwall on the two linking fault segments and subsidence in their hangingwalls. Because of the nature of coastal erosion, the southern Crete study integrates uplift data derived from marine terraces from the footwall of the Ptolemy fault with uplift in the hangingwall of the South Central Crete Fault (SCCF). In this case linking uplift rates and stream behaviour with fault mechanics theory is not straightforward and you need to be very clear and careful in your argumentation.**

*Dr. Ford is correct in that we are trying to assess the response of river profiles to temporal changes in rock uplift rate; however, our object is* not *to "study the timing and stages of fault growth and linkage". Instead, our objective is clearly stated in the final paragraph of the introduction (P. 4, lines 10 – 25). Two sentences that illustrate what our objectives are include: "In this study we seek to determine factors that control the evolution of mountain streams along a rapidly uplifting coastline from the island of Crete, Greece, and to assess the nature in which these streams record tectonic signals and how well they conform to predictions of stream-power incision models (c.f. Tomkins et al., 2003)." (P. 4 lines 10 – 12); and "We exploit fault mechanics theory that predicts a rapid increase in the rate of uplift in the vicinity of a fault linkage to establish a relative uplift history for the study area; first an early phase of uplift*

*consistent with growth of two isolated fault systems followed by a later phase related to the linkage and development of a new composite fault system (Gallen et al., 2014). We then use this natural experiment to assess the fluvial response to this step-change in uplift rate across the fault linkage zone." (P. 4, lines 17 – 21).*

*We apologize for the misunderstanding based on the text in the abstract and have added the following sentence to the abstract to clarify the intent of our study: "We use this natural experiment to assess the response of river profiles to a temporal jump in uplift rate and to assess the applicability of the stream power incision model to this and similar settings."*

**It appears that you are comparing the uplift history of the SCCF hanginwall marine terraces directly to footwall uplift on the Ptolemy fault.**

*Yes, we are comparing the river profile data to all of the independent measurements of local rock uplift rate from marine terraces, which includes the SCCF hanging wall and the Ptolemy fault footwall. This allows us to evaluate assumptions made in river profile analysis. We further use the independent evidence of recent fault linkage to assess the influence of a step change in rock uplift rate on river profile response.*

**A regional component of uplift of the island of Crete appears to be superimposed on vertical motions generated by fault activity, thus exaggerating footwall uplift on both faults (Fig. 1c). When did this regional uplift occur with respect to the fault history?**

*Based on the analysis presented in this study and previous work by Gallen et al. (2014), regional uplift has been active throughout the history of fault development, continues today and likely proceeded fault initiation. Dr. Boulton had a similar comment regarding the relative uplift history as interpreted from river profile analysis. This is something that we address in the Discussion Section 5.3 "Normalized steepness index ($k_{sn}$) patters and uplift history". The last paragraph of section 5.3.1 "Along strike patterns" (P. 19, lines 1 – 12) reads as follows:*

*"In summary, the patterns of $k_{sn}$ for different river reaches reflect a three phase uplift history. Starting with the oldest and moving forward in time, the upper most river reaches above generation-1 knickpoints are interpreted as recording a period of regionally low uplift rate (Fig. 6d). Intermediate river reaches follow an uplift pattern consistent with the pattern of displacement along two independent normal fault systems with a maximum uplift rate near the fault centers that then taper to minima at the fault tips (Fig. 6c). Normalized channel steepness values from the lower footwall reaches are consistent with footwall uplift following linkage of adjacent fault systems, as increased uplift rates occur near the site of fault linkage (Fig. 6b), while $k_{sn}$ values from the hanging wall of the South-Central Crete fault are consistent with lower rates of net (secular + local) rock uplift due to slip on the fault (Fig. 6a). This history is consistent with the conceptual model presented in figure 2 and supports the findings of numerous other studies that show that river profiles are sensitive recorders of relative uplift histories (Snyder et al., 2000; Kirby and Whipple, 2001, 2012; Wobus et al., 2006; Whittaker et al., 2008; Boulton and Whittaker, 2009; Pritchard et al., 2009; Whittaker and Boulton, 2012; Perron and Royden, 2013; Royden and Perron, 2013; Goren et al., 2014; Whittaker and Walker, 2015)."*

**The observed net uplift in the hangingwall of the SCCF indicates that the regional component is greater than the hangingwall subsidence generated by the SCCF as the authors state in page 5, lines 30-33. So it must be a pretty substantial vertical motion! Can you quantify the regional uplift? This regional component should be removed before calculating D on the faults (see below).**

*We agree that there exists substantial regional vertical motion despite active extension in the upper crust. We have quantified the amount of vertical displacement in the absence of faulting and wrote a paper about it that was published in Earth and Planetary Science Letters in 2014 entitled "Active simultaneous uplift and margin-normal extension in a forearc high, Crete, Greece". It is cited several times in this manuscript.*

*The regional signal of uplift does not need to be removed before calculating D because it is similarly effecting footwalls and hanging walls and D is a relative measure with respect to a specific datum, typically the fault trace. We address this comment in more detail below. If one were to remove the regional signal of uplift from the footwalls and hanging walls, the results would be the same as we have presented them.*

**Page 1, Line 16-17. 'Fault mechanics predicts that when adjacent faults link into a single fault the uplift rate in the linkage zone will increase rapidly'. You need to specify that you are referring to footwall uplift. To improve clarity in your analyses of uplift a suggestion is to consistently specify whether you are referring to composite footwall uplift or the composite hangingwall uplift.**

*This is an excellent point. We have revised the above mentioned sentence to read as follows: "Fault mechanics predicts that when adjacent faults link into a single fault the uplift rate in footwalls of the linkage zone will increase rapidly."*

**The data represented in Figure 1c (maximum footwall topography) is derived from the footwall of both faults and shows the classic bell shaped curves typical of two fault segments. As the profiles represent the footwall of both faults the SCCF would not occur between the two profiles as indicated on the graph. It is notable that the footwall uplift on the SCCF is twice that on the Ptolemy fault, implying that it a larger fault, with perhaps a longer history? The highest topography is further back from the fault also suggesting a longer history of uplift and erosion? Can you comment on this? Is the difference in fault size and age relevant for your interpretation of the river behaviour?**

*This is an interesting point, but is difficult to evaluate from topographic data alone. First, the Ptolemy fault is below sea level and is thus much longer than the coastline of the Ptolemy footwall (see figure 1a-inset). If the displacement on the Ptolmey fault is measured from the bathymetric trace of the fault D is approximately 2.5 km, similar to the SCCF footwall. However, using the total displacement as a proxy for fault age can be problematic as it assumes that the fault slip rates are the same, which may not be the case. The age of the fault is not important to our interpretations because the response time of the rivers is likely shorter than the age of the fault. Furthermore, we are only concerned with the more recent evolution of the fault system, specifically the transition from independent faults into a single linked structure. Because such an interpretation relies on many assumptions and is not central to the objectives of this study, we have respectfully decided to exclude discussion of this topic in the present manuscript.*

**The data in figures 1d (marine terrace correlations) and 3b (uplift rate from marine terraces) are derived from the footwall of the Ptolemy fault and the hangingwall of the South Central Crete Fault. The history of footwall uplift can therefore only be deciphered for the Ptolemy Fault. It is on this fault that you see the clear acceleration in uplift toward the relay zone. I am intrigued by the abrupt eastward termination of these footwall terraces along a NS line inland from stations 6-7. Can you comment on this? Is there a N-S fault here or is the map incomplete? This could be relevant for your interpretation.**

*The problem with the mapped extent of the terraces in figure 1b is an oversight on our part. The terraces are not preserved beyond this point, but we interpret that they terminate at the fault scarp. We have adjusted figure 6b to show that the terraces are inferred to stop at the fault trace.*

**The mismatch and incoherence in marine terrace behaviour between the two blocks that you discuss as unusual, is in my opinion, because you are passing eastward across the fault from uplifting footwall to subsiding hangingwall. The decrease in uplift toward the centre of the Dikti block is exactly what you would expect in the subsidence profile of a normal fault hangingwall (Fg. 3b). This also fits perfectly with the fault mechanics model predictions when combined with a regional uplift.**

*We are confused by what the reviewer is referring to, as we do not interpret the mismatch and incoherence in the marine terrace behavior between the two blocks as unusual. We agree with the reviewer's interpretation. We describe our interpretation explicitly in section 2.2 "Tectonic geomorphology of south-central Crete", which is based on Gallen et al. (2014). The latter half of section 2.2. is as follows:*

*"The eastern portion of the Asterousia Mountains is uplifting at eastward-increasing rates between 0.4-0.8 m kyr$^{-1}$ during the Pleistocene as part of the footwall of an offshore normal fault, known as the Ptolemy fault (Fig. 1). At the town of Tsoutsouros, the marine terraces are truncated by a major normal fault that extends on shore, known as the South-Central Crete fault (SCCF), which represents the onshore continuation of the Ptolemy fault (Fig. 1). East of Tsoutsouros the SCCF forms the onshore segmented range front of the Dikti Mountains. Pleistocene marine terraces offset by the SCCF at Tsoutsouros provide long-term average slip rates of ~ 0.35 m kyr$^{-1}$, while the terraces cut into the hanging wall of the SCCF record uplift rates of 0.1 − 0.3 m kyr$^{-1}$ (Gallen et al., 2014). The observation that terraces are uplifted in the hanging wall of the SCCF is of regional importance and signifies that geodynamic processes responsible for regional uplift of the island of Crete are outpacing upper crustal thinning accommodated by motion on active extensional faults. Gallen et al. (2014) further interpret the Ptolemy and SCCF faults as previously independent growing fault systems that linked in the geologically recent past (< 1 Ma) and that the Tsoutsouros area represents the linkage zone between the two fault systems. This interpretation is based on the observations that the long-term displacement history inferred from swath topographic profiles that are parallel to the footwall mountain ranges are consistent with displacement on two isolated fault systems (Fig. 1a, c), while marine terraces that are inferred to have formed at ~ ≤ 400 kyrs are now offset by a single linked fault (Fig. 1d)."*

**P5, section 2.2. I see no evidence for horst structures on the maps you present, ie. I cannot see the conjugate north dipping faults on the north side of the Asterousia and Dikti mountains that would be necessary for these to be horst structures. Your maps indicate that these mountains represent fault block crests in the footwalls of the two studied faults. This would indeed have to be the case for you to apply the fault growth model presented in Figure 2.**

*We apologize for this oversight. The active normal faults on the north side of the Asterousia are well documented and we have added these structures to figure 1. Normal faults are present on the north side of the Dikti Mountains as well, but are less well studied. The spatial extent of the maps shown in figure 1 do not cover the faults on the north side of the Dikti and are thus not included in this figure.*

**Page 13, 4.1: Section 5.1 Calculation of D/L ratio (figure 1). You use maximum topography in the footwall of the two faults as a proxy for displacement and this to calculate D/L ratios for the faults. However observed footwall relief represents only footwall uplift, here combined with the regional uplift component. The throw or vertical component of fault displacement is the sum of footwall uplift**

**plus hangingwall subsidence, a value that you probably don't have, as is often the case. The ratio of long-term footwall uplift to hangingwall subsidence (U:S) has been estimated in various rifts (e.g. Basin and Range; Stein et al 1988) and by elastic dislocation modelling of uplifted terraces in rifts such as the Gulf of Corinth (King, 1998; Armijo et al. 1996). These studies propose U:S values ranging from 1:6 to 1:1 (see McNeill and Collier 2004, for example). If no local constraints are available most authors use values for U:S of 1:2 or 1:3 to derive a reasonable estimate of throw. The displacement along the fault is then calculated assuming a reasonable dip for the normal fault. However you have the additional problem of having to remove the regional uplift component before doing this calculation. Why is D for the Ptolmey fault measured with respect to sea level while D for the South Central Crete fault is measured with respect to a higher reference level (200m above sea level?)?**

*There is a lot to address in this comment, so we will touch on one topic at a time. However, it is important to note that we detail our procedure and terminology in section 2.4 "Relationships between fault growth, linkage and footwall uplift patterns". Furthermore, anyone familiar with the literature described in section 2.2 will be aware of many of the details laid out below.*

*The basis for our D/L measurements is derived from the seminal work on this topic by Dawers et al. (1993). We are effectively calculating a throw measurement with respect to some datum, which is typically taken as the fault trace or some other horizontal datum. This is traditionally how these measurements are made, so we are simply using the same terminology and procedures as previous researchers.*

*Dawers et al. (1993), demonstrated that because the displacement profiles of normal faults are self-similar, D/L ratios will not change so long as they are measured from a common horizontal reference datum. This is important because it means that so long as one measures the length and displacement from the same horizontal reference datum, the D/L measurements will be the same for a given fault. The reference datum is typically chosen as approximately the elevation of the fault trace, but sea level will work just as well. Case in point, the D/L ratio of the Ptolmey fault taking at sea level is ~3.5%, while if it is measured from the bathymetric trace, where D is approximately 2.5 km and L is about 70 km, the D/L ratio is similar, 3.6%.*

*Furthermore, this means that so long as there isn't differential regional uplift across the footwall and hangingwall sides of a fault the D/L ratio measured from a common horizontal reference will remain unaffected. In other words, because this is a relative measure and we have no evidence to indicate that the regional signal of uplift varies over the spatial scale considered in this study, the regional uplift will not affect out D/L measurements. We measure the length and displacement of the SCCF from the fault trace (~200 m above sea level) because we don't know what the fault length is at sea level because it is below the surface.*

*The reviewer seems to be conflating displacement-to-length ratios with ratios of footwall uplift to hanging wall subsidence in part of this comment. We are basing our comparison of D/L ratios to the global compilation of such measurement in Schlische et al. (1996). At any rate, in Gallen et al. (2014) we did calculate the ratio of footwall uplift to hanging wall subsidence from our marine terrace survey as ~ 3:1.*

**As regards the river profile analysis, it is important to emphasise in your interpretations that the western rivers cut across only the footwall of the Ptolemy fault while the eastern rivers cut across both the footwall and hangingwall of the SCC fault. The regional component of uplift should not be forgotten in the interpretation of these rivers.**

*We agree and are careful to identify when we are referring to the footwall or hanging wall of the South Central Crete fault. For example, see figures 5, 6 and 10 and their corresponding captions.*

*We also agree that one should not ignore the regional uplift signal and we do not ignore the regional uplift signal in the study. The regional signal of uplift should effect our entire study area similarly based on the spatial scale that we are investigating. The majority of the interesting spatial and temporal changes in uplift rate are reflected nicely in the river profile geometries, as noted in figures 4-6, and these are best explained by motion on the Ptolemy and South Central Crete faults. If the regional signal of uplift were to perturb the river profiles this would be detected in the river profiles, unless obscured by the changes in uplift rate associated with faulting.*

**I conclude by requesting that the authors clarify the complexity of the structural framework of their study area, the true structural position of data sets, and the terminology used. This will probably have an impact on the presentation of interpretation of tectonic signals in river profiles.**

*We thank Dr. Ford for taking the time to review our manuscript and would like to point out that many of the comments/criticisms that were raised were already addressed and discussed in the text. Furthermore, many of the details on the structural geology are provided in our previous study of Crete by Gallen et al. (2014). We dedicate a great deal of text summarizing the details of that investigation because if forms the foundation for this study. In the end, it is our opinion that the comments provided by Dr. Ford only lead to minor, clarifying modifications to the figures and text and do not require reanalysis or interpretation of our data; our conclusions still hold.*